# INTERPGNN: UNDERSTAND AND IMPROVE GENERALIZATION ABILITY OF TRANSDUCTIVE GNNS THROUGH THE LENS OF INTERPLAY BETWEEN TRAIN AND TEST NODES

**Jiawei Sun**[1], **Kailai Li**[1], **Ruoxin Chen**[4], **Jie Li**[123][*] **Chentao Wu**[1], **Yue Ding**[1], **Junchi Yan**[12]

[1]Department of Computer Science and Engineering, Shanghai Jiao Tong University
[2]MoE Key Lab of AI, Shanghai Jiao Tong University
[3]Yancheng Blockchain Research Institute
[4]Youtu Lab, Tencent
{noelsjw,kailai_li,lijiecs,wuct,dingyue,yanjunchi}@sjtu.edu.cn
cusmochen@tencent.com

## ABSTRACT

Transductive node prediction has been a popular learning setting in Graph Neural Networks (GNNs). It has been widely observed that the shortage of information flow between the distant nodes and out-of-batch nodes (for large-scale graphs) often hurt the generalization of GNNs which overwhelmingly adopt message-passing. Yet there is still no formal and direct theoretical results to quantitatively capture the underlying mechanism, despite the recent advance in both theoretical and empirical studies for GNN's generalization ability. In this paper, the *L-hop interplay* (i.e., message passing capability with training nodes) for an $L$-layer GNN is successfully incorporated in our derived PAC-Bayesian bound for GNNs in the semi-supervised transductive setting. In other words, we quantitatively show how the interplay between training and testing sets influence the generalization ability which also partly explains the effectiveness of some existing empirical methods for enhancing generalization. Based on this result, we further design a plug-and-play **Graph Global Workspace** module for GNNs (InterpGNN-GW) to enhance the interplay, utilizing the key-value attention mechanism to summarize crucial nodes' embeddings into memory and broadcast the memory to all nodes, in contrast to the pairwise attention scheme in previous graph transformers. Extensive experiments on both small-scale and large-scale graph datasets validate the effectiveness of our theory and approaches.

## 1 INTRODUCTION

Graph Neural Networks (GNNs) have shown success in learning both node and graph representations, with applications in node/graph classification (Kipf & Welling, 2017; Veličković et al., 2018; Xu et al., 2019), link prediction (Du et al., 2019b; Yang et al., 2022), etc. Current GNNs mainly use the message-passing paradigm (Gilmer et al., 2017), updating nodes at each layer by aggregating their embeddings with neighboring nodes'. In particular, recent empirical studies show that the effective interplay between train and test nodes either from the raw structure (Lukovnikov & Fischer, 2021; Rusch et al., 2022) or by implicit connection e.g., graph transformers (Wu et al., 2022; 2023) can effectively boost the generalization performance of GNNs for transductive node prediction which is the common setting in existing GNNs and is also the focus of this paper. Yet there still lacks quantitative results on the impact of graph structure especially that between train and test nodes on GNNs' generalization ability, despite the abundant studies (Garg et al., 2020; Scarselli et al., 2018; Garg et al., 2020; Liao et al., 2021; Du et al., 2019a) on theoretical GNN generalization error using different tools e.g., Rademacher complexity, VC dimension, and PAC-Bayesian. Table 1 shows that previous derived bounds are related to different graph-specific information for the transductive setting, including graph diffusion (Esser et al., 2021), disparity in aggregated node

---

[*]Corresponding author

Table 1: Massage-passing based GNNs' generalization error bounds. Note that we are the first to establish the generalization error via the structure interplay between training and test sets. In fact, the interplay has been empirically verified as an effective way for improving generalization in Lukovnikov & Fischer (2021); Rusch et al. (2022).

| Works | Analysis framework | Setting | i.i.d.[1] | Task | Graph-specific information |
|---|---|---|---|---|---|
| Scarselli et al. (2018) | VC dimension | Inductive | ✓ | binary node cls. | number of nodes |
| Verma & Zhang (2019) | algorithmic stability | Inductive | ✓ | multiclass node cls. | eigenvalue of graph filter |
| Garg et al. (2020) | Rademacher complexity | Inductive | ✓ | binary graph cls. | maximum node degree |
| Oono & Suzuki (2020) | Rademacher complexity | Transductive | ✗ | binary node cls. | none |
| Esser et al. (2021) | Rademacher complexity | Transductive | ✓ | binary node cls. | graph diffusion |
| Du et al. (2019a) | Neural Tangent Kernel | Inductive | ✓ | binary graph cls. | none |
| Liao et al. (2021) | PAC-Bayesian | Inductive | ✓ | binary graph cls. | maximum node degree |
| Ma et al. (2021) | PAC-Bayesian | Transductive | ✗ | multiclass node cls. | disparity in aggregated features |
| Maskey et al. (2022) | Convergence speed | Inductive | ✓ | multiclass graph cls. | average number of nodes |
| InterpGNN (Ours) | PAC-Bayesian | Transductive | ✗ | multiclass node cls. | interplay |

[1] i.i.d. means examples are drawn from an identical and independent distribution.

feature (Ma et al., 2021), yet the results concerning graph structure is missing and cannot support the empirical finding on its importance in Lukovnikov & Fischer (2021); Wu et al. (2022).

**Theory perspective:** we derive a generalization error bound for GNNs (Theorem 1 with details in Section 2.2) using the tool of PAC-Bayesian (McAllester, 1998). Its novelty lies in the direct connection to the interplay between the train and test nodes (as depicted by Eq. 8). Specifically, for an $L$-layer GNN, we define $l$-*hop* ($l \leq L$) *interplay* $\mathcal{I}^l$ (formally in Definition 1) as the proportion of test nodes that can interact with training nodes after $l$-times message passing, i.e., those test nodes whose minimum path length to any training node is $l$. As summarized in Table 1, to our best knowledge, our derived bound is the first to directly show that more information interaction between neighboring test and training nodes can lead to smaller generalization error and interpret the structural imbalance phenomenon for homophilic graphs.

Informed by the above theory, increasing the interaction between distant training and test nodes can reduce generalization error. One direct way is deepening the GNN. This is because a GNN with $L$ sequentially stacked layers can utilize up to $L$-hop neighbors for node feature updates. However, most GNNs achieve peak performance with shallow depth, and their performance degrades with increasing layers due to common issues like over-smoothing (Chen et al., 2020a) and over-squashing (Alon & Yahav, 2021). Another way is implicit connections, such as graph transformers (Wu et al., 2023) assign attention scores as the weights of implicit edges between any pair of nodes. The challenge is further pronounced with the fact that current scalable GNNs (Chiang et al., 2019; Zeng et al., 2020) mainly employ mini-batch training protocol, where nodes from different batches are isolated.

**Technical perspective:** we propose a plug-and-play module called Graph **G**lobal **W**orkspace (InterpGNN-GW) for GNNs. The so-called global workspace is composed of memory slots that can store the aggregated embeddings of important nodes. The raw node features are first transformed into embeddings using any GNN (Kipf & Welling, 2017; Hamilton et al., 2017), and random-walk-based techniques like node2vec (Grover & Leskovec, 2016) serve as positional encoding for nodes. Key-value attention is then used to select important node embeddings to write into the workspace , leveraging topological information in positional encoding for coordination across the entire graph. Memory updates employ gated approach to achieve persistence. Finally, memory contents stored in the global workspace are broadcast to all nodes. Compared to the graph transformers (Wu et al., 2022; 2023) based on inner-batch pair-wise attention, the global workspace can provide global consistency. **The contributions are as follows:**

**1) New theory:** We derive a PAC-Bayesian bound for message-passing GNN in the transductive node classification, which is the dominant setting for graph learning. The bound associates the generalization error with the topological interplay between training and test sets, i.e., whether test nodes can obtain the features of training nodes through message passing. The bound theoretically interpret the structure imbalance phenomenon observed in transductive node classification.

**2) Plug-in method:** Based on the derived bound, we propose the Graph Global Workspace (InterpGNN-GW) to enhance the interplay between distant and out-of-batch nodes. Nodes with positional encoding compete to write to and read from the global workspace via key-value attention.

**3) Strong results:** Experiments on small- and large-scale graph datasets show InterpGNN-GW's superior performance on node classification compared to scalable GNNs and graph transformers.

## 2 GENERALIZATION ERROR BOUND VIA PAC-BAYESIAN ANALYSIS

### 2.1 PROBLEM SETUP AND PRELIMINARIES ON PAC-BAYESIAN

**Transductive node classification.** We consider semi-supervised transductive $K$-class node classification which is commonly studied in the graph learning community. Let $G = (V, E)$ represent an undirected graph, where $V$ denote the node set with size $N$, and $E$ denote the edge set. Let $\mathbf{X}$ denote the node features, $Y$ denote the node labels and $\mathbf{A}$ represent the adjacent matrix. Assume that for each node $i$, its label is generated by a true but unknown distribution $D$ conditioned on node features and neighbors. Denote the set of training nodes as $S$ and the set of test nodes as $U$. In the transductive node classification task, a classifier $f(\mathbf{A}, \mathbf{X})$ is learned using the labels of training set $Y_S$, the features of both training and test nodes $\mathbf{X} = [\mathbf{X}_S, \mathbf{X}_U]$, and the adjacency matrix $\mathbf{A}$. The objective is to classify the unlabeled nodes in $U$. The classification of the $i$-th node is:

$$\hat{y}_i = \underset{k \in \{1, \cdots K\}}{\arg\max} f(\mathbf{A}, \mathbf{X})[i, k],$$

where $f(\mathbf{A}, \mathbf{X})[i, k]$ means the $(i, k)$-th element of the obtained matrix.

$\gamma$**-margin ramp loss.** We use the $\gamma$-margin ramp loss following Bartlett et al. (2017); Zhang et al. (2019); Koltchinskii & Panchenko (2002). First define the *margin operator* of a hypothesis $f$ for a labeled node as $\mathcal{M}_f(i, y_i) = f(\mathbf{A}, \mathbf{X})[i, y_i] - \max_{k \neq y_i} f(\mathbf{A}, \mathbf{X})[i, k]$. Given the set of training nodes $S$, the *empirical error* of $f$ on $S$ is defined as:

$$R_{S,\gamma}(f) := \frac{1}{|S|} \sum_{i \in S} \Phi^\gamma(\mathcal{M}_f(i, y_i)), \tag{1}$$

where the ramp loss $\Phi^\gamma(\cdot)$ is defined as:

$$\Phi^\gamma(a) := \begin{cases} 0 & \gamma \leq a, \\ 1 - a/\gamma & 0 \leq a \leq \gamma, \\ 1 & a \leq 0. \end{cases} \tag{2}$$

Additionally, we use $\Phi^0(a) = \mathbb{1}[\hat{y}_i \neq y]$ to denote the 0-1 classification loss and its corresponding empirical error $R_{S,0}(f)$, since the marginal ramp loss transitions to classification loss as $\gamma$ approaches zero. Similarly we can define the empirical error on the test set $U$ as $R_{U,\gamma}$. The *excepted error* on training and test set are defined as:

$$r_{S,\gamma}(f) := \underset{y \sim D}{\mathbb{E}} [R_{S,\gamma}(f)] \quad r_{U,\gamma}(f) := \underset{y \sim D}{\mathbb{E}} [R_{U,\gamma}(f)], \tag{3}$$

**PAC-Bayesian background.** PAC-Bayesian (McAllester, 1998; 1999) provides a tool to analyze the generalization performance of randomized predictors. Consider a training set $S$, a true yet unknown label-generating distribution $D$ from which samples are drawn, a hypothesis class $\mathcal{H}$ of classifiers, a *prior* distribution $P$ over $\mathcal{H}$, and a *posterior* distribution $Q$ over $\mathcal{H}$. The prior $P$ encodes our *prior* about the model before seeing any training data; the *posterior* is learned by learning on training data. As opposed to common learning paradigm which learns a single deterministic predictor, PAC-Bayesian studies the distribution of models, and the error measure on test set is extended as,

$$R_{U,\gamma}(Q) = \underset{f \sim Q}{\mathbb{E}}[R_{U,\gamma}(f)], \quad r_{U,\gamma}(Q) = \underset{f \sim Q}{\mathbb{E}}[r_{U,\gamma}(f)]. \tag{4}$$

The error measurement on the training set uses a similar definition for $R_{S,\gamma}(Q)$ and $r_{S,\gamma}(Q)$. The aim of PAC-Bayesian is to bound the gap between $R_{S,\gamma}(Q)$ and $r_{U,\gamma}(Q)$ with high probability. The PAC-Bayesian typically first associates the generalization bound with the discrepancy between any possible prior $P$ and posterior $Q$, measured with the KL divergence $\mathcal{D}_{KL}(Q||P)$. Then the concrete bounds are obtained by constructing specific priors and posteriors. Although PAC-Bayes is generally founded on stochastic models, McAllester (2003b); Neyshabur et al. (2017) provide standard techniques for obtaining the generalization bounds of deterministic models. We refer readers to tutorials (Guedj, 2019; Alquier, 2021) for more details on PAC-Bayesian.

### 2.2 THE DERIVED STRUCTURE-AWARE PAC-BAYESIAN BOUND OF GCN

**PAC-Bayesian bound for Transductive Learning.** The following lemma with proof in Appendix B.1 introduces PAC-Bayesian bound for general transductive learning.

**Lemma 1 (PAC-Bayesian bound for transductive learning)** *Given a full set $V$ with $N$ examples, for any training set $S$ of size $m$ and test set $U$, for any prior distribution $P$ on the hypothesis space of classifier $\mathcal{H}$ (not necessarily a GNN), for any $\delta \in (0,1]$, with probability at least $1 - \delta$, we have,*

$$r_{U,\gamma}(Q) \le R_{S,\gamma}(Q) + \sqrt{\frac{\mathcal{D}_{KL}(Q||P) + \ln\frac{2m}{\delta} + D(P)}{2(m-1)}} \tag{5}$$

where $D(P) = \ln \mathbb{E}_{f \sim P}\left[e^{2(m-1)(r_{U,\gamma}(f) - r_{S,\gamma}(f))^2}\right]$ quantifies the expected loss discrepancy between the training and test set. Lemma 1 is a natural adaptation of PAC-Bayesian in Germain et al. (2009) from inductive learning to transductive learning. Using lemma 1 as a tool for PAC-Bayesian, the generalization bounds can be obtained by upper-bounding the term $D(P)$. For GNNs, given that embedding can be propagated to neighbors, we need to consider the impact of structure on the bound of $D(P)$ in comparison to general transductive learning.

**PAC-Bayesian bound for GCN.** We now incorporate the node correlation induced by the graph structure into Lemma 1. We first denote that the $l_2$-norm of node features are bounded by $B_x$, and the $l_2$-norm of weights for each layer, for any classifier is bounded by $B_w$. We analyze GCN (Kipf & Welling, 2017). The derived bound can be easily generalized to any GNNs following the *message-passing* and *aggregation* paradigm such as GraphSAGE (Hamilton et al., 2017), SGC (Wu et al., 2019). The $l$-th layer of GCN is defined as :

$$\mathbf{H}_l = \sigma_l(\tilde{\mathbf{A}}\mathbf{H}_{l-1}\mathbf{W}_l), \tag{6}$$

where $\sigma_l(\cdot)$ is ReLU non-linear activation function, $\mathbf{W}_l$ is the trainable weight matrix, $\mathbf{H}_l$ are node embeddings of the $l$-th layer with $\mathbf{H}_0 = \mathbf{X}$, $\tilde{\mathbf{A}} = \mathbf{A} + \mathbf{I}$, and $\mathbf{I}$ is the identity matrix. The maximum number of hidden units across all layers is denoted as $h$. A path between node $i$ and node $j$ is denoted as a sequence of successively connected nodes: $p(i,j) = (v_i, \cdots, v_j)$, with the path length $\omega(p(i,j))$ defined as the count of edges along it. The shortest path length is denoted by $s(i,j) = \min\{\omega(p(i,j))\}$. For the first time to our best knowledge, to explore the impact of structural correlation between training and test sets on generalization error, we define the $L$-**hop interplay** between training and test nodes.

**Definition 1 ($L$-hop interplay)** *For a graph $G = (V,E)$ with training set $S \subseteq V$ and test set $U \subseteq V$, the $L$-hop interplay $\mathcal{I}_L$ between $S$ and $U$ is defined as:*

$$\mathcal{I}_L = \frac{\sum_{r=1}^{L}(L - r + 1) \cdot |P_r|}{|S| \cdot |U|}, \tag{7}$$

*where $P_r := \{(i,j)|i \in S, j \in U, s(i,j) = r\}$ represents training-test node pairs with the shortest path length of $r$.*

We now derive the following generalization bounds of $L$-layer GCN by associating the upper bound of $D(P)$ with the $L$-hop interplay. The complete derivation is given in Appendix B.2. The proof is in two steps: 1) bounding $D(P)$ based on the premise that embedding exchange among connected nodes leads to closer output logits, 2) bounding $D_{KL}(Q||P)$ as per Neyshabur et al. (2017); Ma et al. (2021).

**Theorem 1 (PAC-bayes bounds with $L$-hop interplay for transductive GCN)** *Let $f \in \mathcal{H}$ be an $L$-layer GCN with parameters $\{\mathbf{W}_i\}_{i=1}^{L}$, for any $B_x, B_w > 0$, $L, h, K \ge 1$, and any $\delta, \gamma > 0$, with probability at least $1 - \delta$ over a training set $S$ of size $m$ we have,*

$$r_{U,0}(f) \le R_{S,\gamma}(f) + O\left(\sqrt{\frac{B_x^2 L^2 h \ln(4Lh)\Pi\Sigma + \ln\frac{Lm}{\delta}}{\gamma^2 m} + K^2 B_x^2 (1 - \frac{2\mathcal{I}_L}{d_{max}})^2 B_w^L}\right) \tag{8}$$

*where $\Pi = \prod_{l=1}^{L} \|\mathbf{W}_l\|_F^2$ and $\Sigma = \sum_{l=1}^{L} \frac{\|\mathbf{W}_l\|_F^2}{\|\mathbf{W}_l\|_2^2}$ are product and sum of spectral norm of weights, $d_{max}$ is the maximum node degree.*

**Theorem 1 interpret the structure imbalance phenomenon.** Grouping test nodes based on their distance with training nodes, the impact of topological structure on generalization is presented in Fig. 1 with further elaboration in Appendix C. For test nodes exhibiting stronger interplay with training nodes, the generalization error is smaller. Theorem 1 theoretically interprets this experimental observation. Considering Tang et al. (2020); Liu et al. (2023) connect generalization ability with node degree, we modify the grouping criterion to node degree, revealing a similar trend (Fig. 2). We attribute this to nodes with smaller degrees engage in fewer interactions with training nodes. The generalization of above theorem on heterophilic graphs please see in Appendix G.3

## 3 ENHANCING GENERALIZATION VIA THE GRAPH GLOBAL WORKSPACE

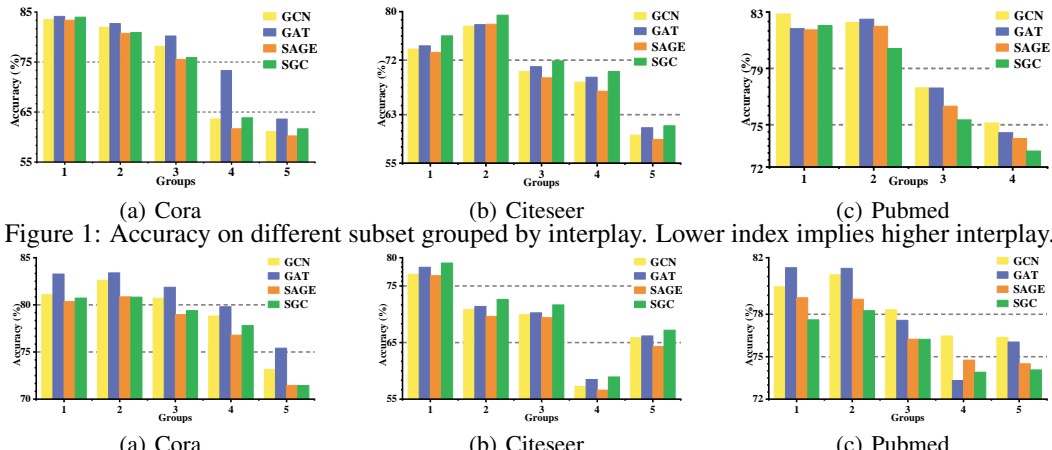

(a) Cora        (b) Citeseer        (c) Pubmed

Figure 1: Accuracy on different subset grouped by interplay. Lower index implies higher interplay.

(a) Cora        (b) Citeseer        (c) Pubmed

Figure 2: Accuracy on different subset grouped by nodes degree. Lower index implies higher degree.

**Intuition.** Building upon previous investigations, exploring ways to enhance the interplay between distant nodes is worth researching. We propose a plug-and-play module for GNNs, inspired by the Global Workspace Theory (Baars, 2005; VanRullen & Kanai, 2021; Goyal et al., 2022), to enhance long-range node interplay and reduce complexity. Nodes use the key-value attention mechanism to competitively write their embeddings into the workspace, and subsequently retrieve information from the workspace. Distant test nodes can indirectly interplay with training nodes through information stored in the workspace. For large-scale graphs trained in mini-batch, interplay among out-of-batch nodes is feasible. This section elucidates the three steps of our method.

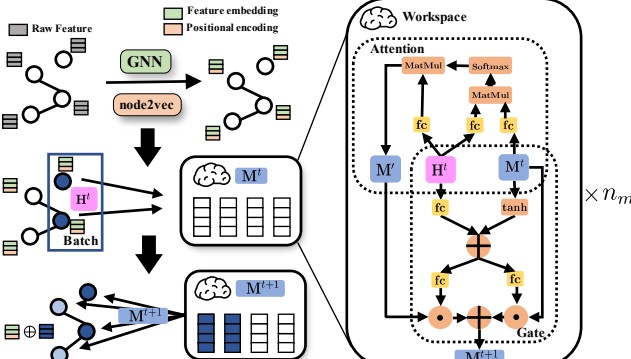

Figure 3: InterpGNN-GW consists of three steps: 1) extract initial feature embedding and positional encoding. 2) Nodes compete for writing access to the global workspace. 3) Nodes retrieve information from the global workspace. The right part shows step two, where attention selects nodes and updates $n_m$ memory slots using the gate mechanism, with $H^t$ and $M^t$ as current node embeddings and memory.

**Initial feature embedding and positional encoding.** We first employ a graph neural network to encode raw node features into $n_f$-dimensional feature embeddings. For a small-scale graph $G = (V, E, \mathbf{A})$ with node feature matrix $\mathbf{X}$, we employ a single-layer GCN to initially encode the feature matrix to a hidden embedding matrix $\mathbf{H} = \sigma(\mathbf{AXW}_0)$. For a large-scale graph, we adopt the mini-batch training method. We use the graph clustering method such as METIS (Karypis & Kumar, 1998) to partition $G$ into $C$ subgraphs $\{G_c = [V_c, E_c]\}_{c=1}^C$, where $V_c$ is the node set of the $c$-th subgraph, and $E_c$ contains edges among nodes in $V_c$. The feature matrix $\mathbf{X}$ and labels $Y$ can also be partitioned into $\{\mathbf{X}_c\}_{c=1}^C$ and $\{Y_c\}_{c=1}^C$. In each epoch, we randomly select a subgraph $G_c$ and obtain the embeddings for nodes in $V_c$ with $\mathbf{H}_c = \sigma(\mathbf{A}_c \mathbf{X}_c \mathbf{W}_0)$. The GCN encoder can be replaced with other GNNs such as GAT (Veličković et al., 2018), GAS (Fey et al., 2021). Node2vec is trained on the entire graph, producing $n_p$-dimensional embedding employed as positional encoding. The node2vec embeddings capture the structural properties of the graph by assigning similar embeddings to close nodes. The positional encoding and clusters are generated only with the adjacency matrix which is accessible during training for transductive learning. For the rest of this section, we focus on the approach for large-scale graphs since full-batch training on small-scale graph can be considered as having a single partitioned subgraph ($C = 1$).

**Nodes compete to write into the workspace.** The workspace selects important nodes via attention mechanisms and stores their embeddings. The global workspace is composed of $n_m$ memory slots,

each of which is a vector of length $l_m$, denoted as $\mathbf{M} = [m_1, \cdots, m_{n_m}]$. *Keys* and *Values* are obtained by multiplying node embeddings with two different linear layers, i.e., $\mathbf{K}_w = \mathbf{H}_c \mathbf{W}_{w,k}$ and $\mathbf{V}_w = \mathbf{H}_c \mathbf{W}_{w,v}$. Similarly, we can obtain the *Query* matrix with memory as $\mathbf{Q}_w = \mathbf{M}^t \mathbf{W}_{w,q}$. The attention scores of the workspace w.r.t. the nodes are obtained by applying Softmax to the dot-product of Query and Key. The product of the attentions scores and Value updates the workspace,

$$\mathbf{M}^{t+1} \leftarrow \text{softmax}\left(\frac{\mathbf{Q}_w \mathbf{K}_w^\top}{\sqrt{d_k}}\right) \mathbf{V}_w = \text{softmax}\left(\frac{\mathbf{M}^t \mathbf{W}_{w,q} (\mathbf{H}_c \mathbf{W}_{w,k})^\top}{\sqrt{d_k}}\right) \mathbf{H}_c \mathbf{W}_{w,v}. \quad (9)$$

The gated approach (Santoro et al., 2018) is used to update the workspace for dynamic persistence.

**Nodes reload information from the workspace.** Utilizing the attention mechanisms again, the information within the workspace is broadcast to all nodes in $V_c$ to update their embeddings. In this step, node embeddings construct queries $\mathbf{Q}_r = \mathbf{H}_c \mathbf{W}_{r,q}$, and memory slots are utilized to build keys and values, $\mathbf{K}_r = \mathbf{M}^{t+1} \mathbf{W}_{r,k}, \mathbf{V}_r = \mathbf{M}^{t+1} \mathbf{W}_{r,k}$. The attention score of nodes on each memory slots is calculated by $\text{softmax}(\frac{\mathbf{Q}_r \mathbf{K}_r^\top}{\sqrt{d_k}})$. Thus, the information received by each node from all memory slots weighted by attention score is:

$$\widehat{\mathbf{H}}_c = \text{softmax}\left(\frac{\mathbf{H}_c \mathbf{W}_{r,q} (\mathbf{M}^{t+1} \mathbf{W}_{r,k})}{\sqrt{d_k}}\right) \mathbf{M}^{t+1} \mathbf{W}_{r,v}. \quad (10)$$

The embeddings from the workspace contain partial information from all subgraphs. After incorporating the information reloaded from the workspace into the node embeddings, we perform an additional message-passing to obtain the final node embeddings:

$$\mathbf{H}_o = \sigma(\mathbf{A}_\mathbf{c}(\widehat{\mathbf{H}}_c \oplus \mathbf{H}_c)\mathbf{W}_o), \quad (11)$$

where the $\oplus$ can be tensor addition or concatenation operations. In the next epoch, a new subgraph is drawn and the process returns to the first step. This procedure iterates until the model converges or reaches the maximum training epochs. The training algorithm is described in Appendix E.

## 4 EXPERIMENTS

### 4.1 PROTOCOL AND SETUPS

**Task and datasets.** We conduct experiments on semi-supervised transductive node classification on both small-scale and large-scale graphs, adopting full-batch and mini-batch training, respectively. The five small graph datasets include: CoraFull, Wiki-CS, AmazonPhoto, AmazonComputer, and Flickr, and the three large-scale graphs are AMiner-CS, Reddit, Amazon2M. Details are in Appendix F.1. We use a 60%/20%/20% train/validation/test random split for all datasets.

**Implementation details of our approach.** Firstly, we utilize GCN to encode the initial node features into embeddings with dimension $n_d$, which is selected from $\{32, 64, 128, 256\}$. The dimension of the positional encoding is selected from $\{32, 128\}$. For the global workspace module, the number of memory slots is selected from $\{8, 16, 32\}$, and their dimensions can be opted within range of 32-1024. A multi-head dot product attention mechanism is utilized to govern the reading and writing of node embeddings within the global workspace. The above implementation is denoted as *InterpGCN-GW*. We additionally use the historical embeddings in GNNAutoScale (Fey et al., 2021) and denote this implementation as *InterpGAS-GW*. We repeat the experiment for 10 times with different initialization and record the mean and standard deviation. All experiments are conducted on a Titan 2080Ti with 11GB memory. More implementation details are in Appendix F.2.

**Baselines.** We compare three groups of GNNs: 1) **Full-batch GNNs**: GCN (Kipf & Welling, 2017), GAT (Veličković et al., 2018), APPNP (Klicpera et al., 2019). 2) **Scalable GNNs**: ClusterGCN (Chiang et al., 2019), GraphSAINT (Zeng et al., 2020), GRAND+ (Feng et al., 2022), and GNNAutoScale (Fey et al., 2021). 3) **(Scalable) graph transformers**: SAN (Kreuzer et al., 2021), GraphGPS (Rampášek et al., 2022), Graphormer (Ying et al., 2021), and the recent scalable ANS-GT (Zhang et al., 2022b), NAGphormer (Chen et al., 2023) and NodeFormer (Wu et al., 2022). For large graphs, we only compare with scalable GNNs/graph transformers. The reported numbers are quoted from the papers given the same setting otherwise we run authors' code to produce the results.

### 4.2 MAIN RESULTS

**Results on small graphs (ranging from 2k to 100k nodes).** For small graphs, we use full-batch training, and the results are presented in Table 3. InterpGNN-GW outperforms all baselines, which

Table 2: Comparison of scalable GNN and scalable graph transformers for large-scale node classification with mini-batch training. Our approach lowers the $O(|V_c|^2)$ memory complexity of graph transformers' pair-wise attention matrix to $O(|V_c|n_m)$.

| Methods | Mini-batch sample | Time complexity[1] | Memory complexity[1] | Out-of-batch comm. | Interplay enhancement | Positional encoding |
|---|---|---|---|---|---|---|
| ClusterGCN (Chiang et al., 2019) | METIS | $O(|V_c| + |E_c|)$ | $O(|V_c|d + d^2)$ | ✗ | ✗ | ✗ |
| GraphSAINT (Zeng et al., 2020) | Random walk | $O(|V_c| + |E_c|)$ | $O(|V_c|d + d^2)$ | ✗ | ✗ | ✗ |
| Grand+ (Feng et al., 2022) | Random | $O(|V_c| \cdot k \cdot M)$ | $O(|V_c| \cdot k \cdot Md + d^2)$ | ✗ | Random propagation | ✗ |
| GNNAutoScale (Fey et al., 2021) | METIS | $O(|V_c| + |\mathcal{N}_o| + |E_c|)$ | $O(|V_c|d + \mathcal{N}_od + d^2)$ | Historical embed | ✗ | ✗ |
| ANS-GT (Zhang et al., 2022b) | MAB-based | $O(|V|S(n_s + n_g + |V_c|^2))$ | $O((n_s + n_g + |V_c|)^2 + |V_c|d + d^2)$ | ✗ | Pair-wise attention | Proximity encoding |
| NodeFormer (Wu et al., 2022) | Random | $O(|V_c|)$ | $O(|V_c|d^2 + |V_c|d + d^2)$ | ✗ | Pair-wise attention | Node connectivity |
| NAGphormer (Chen et al., 2023) | Hop2token | $O(|V_c|(n_h + 1)^2)$ | $O(|V_c|(n_h)^2 + |V_c|n_hd + d^2)$ | ✗ | Pair-wise attention | Laplacian Eigenvector |
| InterpGNN (Ours) | METIS | $O(|V_c|n_ml_m) + |E_c|$ | $O(|V_c|n_ml_m + |V_c|d + d^2)$ | Global workspace | Store important nodes with attention | Node2vec |

[1] Time and memory for a mini-batch. $G_c = (V_c, E_c)$ is the sampled subgraph for current batch. Assume that both the input and output dimensions of the embeddings are $d$. $n_m$ in our method is the number of memory slots. $n_h$ in NAGphormer denotes the $n_h$-hop neighborhood. $M$ in Grand+ is the number of augmented feature matrices. $n_s$, $n_g$ and $S$ in ANS-GT denote number of super-nodes, global nodes, and augmentations. For memory complexity, $|V_c|n_ml_m$ is on attention matrix, $|V_c|d$ is on node embeddings, and $d^2$ is on linear layers.

Table 3: Node classification on small-scale graphs (% + std). OOM: out of memory. Best in **bold**.

| Datasets | CoraFull | Wiki-CS | Computer | Photo | Flickr |
|---|---|---|---|---|---|
| | | | Full-batch GNNs | | |
| GCN (Kipf & Welling, 2017) | $68.94 \pm 0.23$ | $79.82 \pm 1.02$ | $89.34 \pm 0.62$ | $93.24 \pm 0.27$ | $49.65 \pm 0.15$ |
| GAT (Veličković et al., 2018) | $67.51 \pm 0.19$ | $80.56 \pm 0.82$ | $90.24 \pm 0.25$ | $93.59 \pm 0.33$ | $48.29 \pm 0.21$ |
| APPNP (Klicpera et al., 2019) | $67.01 \pm 0.27$ | $80.98 \pm 0.86$ | $90.81 \pm 0.21$ | $94.51 \pm 0.23$ | $50.12 \pm 0.36$ |
| | | | Scalable GNNs | | |
| ClusterGCN (Chiang et al., 2019) | $69.21 \pm 0.32$ | $81.23 \pm 0.51$ | $90.51 \pm 0.31$ | $94.28 \pm 0.45$ | $48.23 \pm 0.49$ |
| GraphSAINT (Zeng et al., 2020) | $71.32 \pm 0.13$ | $81.65 \pm 0.29$ | $90.93 \pm 0.55$ | $94.94 \pm 0.24$ | $51.01 \pm 0.19$ |
| Grand+ (Feng et al., 2022) | $\mathbf{71.89 \pm 0.40}$ | $82.11 \pm 0.54$ | $89.51 \pm 0.22$ | $94.81 \pm 0.12$ | $51.31 \pm 0.31$ |
| GNNAutoScale (Fey et al., 2021) | $71.08 \pm 0.36$ | $82.78 \pm 0.39$ | $88.92 \pm 0.34$ | $95.68 \pm 0.42$ | $49.31 \pm 0.45$ |
| | | | Scalable GTs | | |
| SAN (Kreuzer et al., 2021) | $66.49 \pm 0.55$ | $78.23 \pm 0.63$ | $89.33 \pm 0.26$ | $93.12 \pm 0.29$ | $50.52 \pm 0.38$ |
| GraphGPS (Rampášek et al., 2022) | $65.62 \pm 0.38$ | $79.35 \pm 0.16$ | OOM | $94.87 \pm 0.45$ | OOM |
| Graphomer (Ying et al., 2021) | OOM | OOM | OOM | $93.61 \pm 0.32$ | OOM |
| NAGphomer (Chen et al., 2023) | $71.51 \pm 0.13$ | $78.62 \pm 0.23$ | $91.22 \pm 0.14$ | $95.49 \pm 0.11$ | $51.71 \pm 0.23$ |
| NodeFormer (Wu et al., 2022) | $65.38 \pm 0.91$ | $80.69 \pm 1.19$ | $88.49 \pm 1.07$ | $93.40 \pm 0.46$ | $49.84 \pm 0.78$ |
| InterpGCN-GW (ours) | $71.72 \pm 0.37$ | $82.92 \pm 0.18$ | $\mathbf{92.10 \pm 0.23}$ | $95.46 \pm 0.15$ | $52.41 \pm 0.22$ |
| InterpGAS-GW (ours) | $71.31 \pm 0.31$ | $\mathbf{83.31 \pm 0.26}$ | $91.27 \pm 0.37$ | $\mathbf{96.13 + 0.20}$ | $\mathbf{53.55 \pm 0.32}$ |

Table 4: Node classification on large graphs (% + std).

| Datasets | Reddit | AMiner-CS | Amazon2M |
|---|---|---|---|
| | Scalable GNNs | | |
| ClusterGCN (Chiang et al., 2019) | $92.69 \pm 0.19$ | $65.73 \pm 0.28$ | $86.32 \pm 0.28$ |
| GraphSAINT (Zeng et al., 2020) | $91.89 \pm 0.17$ | $65.64 \pm 0.49$ | $85.64 \pm 0.14$ |
| Grand+ (Feng et al., 2022) | $92.81 \pm 0.27$ | $66.49 \pm 0.41$ | $86.04 \pm 0.32$ |
| GNNAutoScale (Fey et al., 2021) | $93.11 \pm 0.19$ | $67.11 \pm 0.31$ | $86.32 \pm 0.25$ |
| | Scalable GTs | | |
| ANS-GT (Zhang et al., 2022b) | $90.47 \pm 0.37$ | $65.42 \pm 0.58$ | $87.61 \pm 0.31$ |
| NodeFormer (Wu et al., 2022) | $89.42 \pm 0.28$ | $64.54 \pm 0.35$ | $88.01 \pm 0.29$ |
| NAGphomer (Chen et al., 2023) | $92.87 \pm 0.36$ | $66.92 \pm 0.44$ | $86.20 \pm 0.39$ |
| InterpGCN-GW (ours) | $\mathbf{93.35 \pm 0.20}$ | $68.76 \pm 0.31$ | $\mathbf{89.11 \pm 0.14}$ |
| InterpGAS-GW (ours) | $92.75 \pm 0.23$ | $\mathbf{69.01 \pm 0.22}$ | $88.59 \pm 0.19$ |

Table 5: Non-rigorous inductive node classifications (% + std).

| | CoraFull | Wiki-CS |
|---|---|---|
| ClusterGCN | $64.54 \pm 0.29$ | $75.74 \pm 0.57$ |
| GRAND+ | $67.15 \pm 0.24$ | $77.95 \pm 0.49$ |
| NodeFormer | $65.90 \pm 0.36$ | $76.49 \pm 0.76$ |
| InterpGCN-GW | $68.78 \pm 0.34$ | $81.49 \pm 0.29$ |
| Amazon2M | Aminer-CS | Reddit |
| $83.85 \pm 0.27$ | $61.59 \pm 0.37$ | $89.77 \pm 0.35$ |
| $84.12 \pm 0.18$ | $62.48 \pm 0.36$ | $92.05 \pm 0.23$ |
| $84.25 \pm 0.41$ | $62.12 \pm 0.24$ | $90.75 \pm 0.25$ |
| $88.13 \pm 0.24$ | $67.85 \pm 0.19$ | $92.79 \pm 0.19$ |

can be attributed to the global workspace preserving the important training node embeddings and conveying them to remote test nodes during inference. For scalable graph transformers including NAGphomer and NodeFormer, which utilize the node-pair attention mechanism, InterpGNN-GW surpasses them notably. While the vanilla graph transformers such as GraphGPS, and Graphomer are infeasible on some small graphs due to out-of-memory errors. The introduction of historical embedding on small graphs does not guarantee performance improvement.

**Results on large graphs (ranging from 200k to 2M nodes).** Like ClusterGCN, our methods also use METIS (Karypis & Kumar, 1998) to partition the graph into clusters. For the other two compared baselines GraphSAINT and Grand+, we keep using their own partition method as suggested by the authors. It is worth noting that, InterpGNN-GW outperforms other baselines more notably than in small graph experiments, suggesting its effectiveness in mini-batch training where the message-passing between disparate batches is hindered. While in InterpGNN-GW, nodes from distinct batches can still exchange messages through the shared memory in the global workspace in a persistent way. InterpGAS-GW outperforms InterpGCN-GW on some large graphs, indicating that historical embeddings can enhance the expressiveness of global workspace for mini-batch training. See Appendix G.1 for detailed memory comparison.

## 4.3 FURTHER STUDY AND DISCUSSION

**Results on non-rigorous inductive learning.** In order to further demonstrate the advantages of InterpGNN-GW, we conducted experiments in a non-rigorous inductive setting, where the model during training has access only to the feature of the training nodes, but not feature of the test nodes. Our method is capable of broadcasting the workspace to test nodes, even when these nodes still unseen during the training process. Thus, InterpGNN can efficiently convey information of training

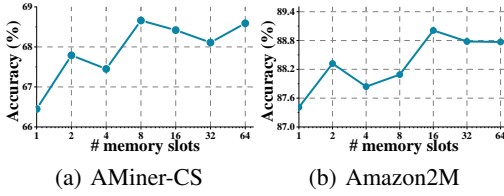
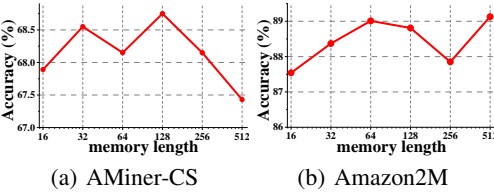

| (a) AMiner-CS | (b) Amazon2M | (a) AMiner-CS | (b) Amazon2M |
|---|---|---|---|

Figure 4: Study on the number of memory slots. Figure 5: Study on the length of memory slots.

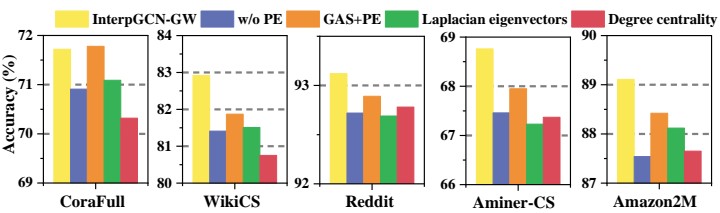
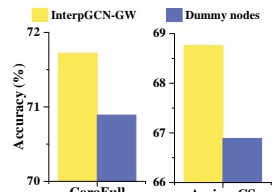

Figure 6: Ablation study on positional encoding.

Figure 7: Comparison with dummy-nodes methods.

Table 6: Comparison between InterpGCN-GW and GNNAutoScale (GAS) Fey et al. (2021) in terms of **accuracy, GPU memory, and average training time per epoch** .

| Batch size | Reddit (#nodes per cluster=39, #edges/#nodes=49.8) | | AMiner-CS (#nodes per clusters=990, #edges/#nodes=10.4) | |
|---|---|---|---|---|
| | InterpGCN-GW | GNNAutoScale | InterpGCN-GW | GNNAutoScale |
| 32 | 91.7%/1561M/5.4s | 91.5%/2537M/6.2s | 64.1/3023M/2.3s | 63.5/2207M/1.4s |
| 64 | 91.6%/1809M/6.1s | 91.8%/3341M/7.6s | 65.4/5391M/2.7s | 64.9/2729M/1.8s |
| 128 | 92.4%/2703M/7.4s | 92.3%/4609M7.5s | 66.4/8503M/3.8s | 65.8/5145M/3.1s |
| 256 | 93.0%/5171M/7.6s | 92.8%/8917M/7.6s | 68.0/9021M/4.5s | 66.6/7543M/4.9s |
| 512 | 93.4%/6989M/8.5s | 93.1%/10485/7.4s | 67.2/10619M/6.6s | 66.5/9325M/7.8s |
| 1024 | 93.5%/8067M/10.8 | OOM | 68.2/10221M/7.8s | 67.1/10613M/8.1s |

nodes to test nodes during the test phase. Unlike other methods that exhibit significant performance degradation in inductive settings compared to transductive ones, InterpGCN-GW experiences only a minor decrease in performance in this non-rigorous inductive setting, closely mirroring its transductive results. The experimental outcomes are detailed in Table 5.

**Ablation on the ability of memory persistence across nodes.** Table 10 shows an ablation study, in which memory slots in the global workspace are initialized in each batch to store only the node information within the current batch. It reveals that removing persistence negatively impacts performance on both large and small graphs, especially for large graphs. We also conducted a comparison on whether to use a gate mechanism when updating the memory, with results in Table 10.

**Ablation on positional encoding.** Table 6 compares the variants without positional encoding or using alternatives e.g. Laplacian eigenvectors in Kreuzer et al. (2021); Chen et al. (2023), and degree centrality in Ying et al. (2021). Removing node2vec positional encoding, which serves as positioning in memory read/write processes, leads to a significant performance drop. The greater gain on large graphs reveals the importance of relative topological information for non-interacting nodes in separate batches. Deterministic positional embeddings calculated directly from the adjacency matrix, are less effective at coordinating nodes than adaptive node2vec embeddings trained on graph structure, especially on large graphs. Additionally, the better performance of InterpGCN-GW compared to GAS+PE indicates the superiority of global workspace over historical embedding.

**Study on number and size of memory slots.** Fig. 4 and Fig. 5 examine the impact of memory slot number ($n_m$) and length ($l_m$) on large-scale datasets. Due to 2080Ti's 11GB memory constraints, we test up to 64 slots. With a relatively small number of memory slots ($n_m \leq 16$), the model's performance improves as $n_m$ increases. Beyond that, further increments in memory slots yield modest performance enhancements. The trend in the length of the memory is similar to the number of memory slots. However, an excessively large memory length decreases the accuracy for Aminer-CS. Meanwhile, the best results for Amazon2M is obtained when the memory size is 512.

**Comparison with GNN variants using historical embeddings.** Our method shows greater improvements on large graphs, as the workspace stores summaries of previous batches. Likewise, GNNAutoScale (Fey et al., 2021) stores historical embeddings in the CPU and transfers out-of-batch

1-hop neighbors' embeddings to the GPU during batch training. We evaluated global workspace and historical embedding efficiency on dense (Reddit) and sparse (AMiner-CS) datasets with varying batch sizes (Table 6). Both methods improve with increasing batch size, but historical embedding occupies more memory on dense datasets and less on sparse ones. Global workspace has two advantages: 1) It requires fixed extra memory, while historical embeddings' GPU memory depends on graph structure, causing more out-of-batch neighbors to be swapped from the CPU in dense datasets. 2) Historical embedding is confined to fetching 1-hop neighbors from the CPU, limiting the enhancement of long-distance node interplay.

**Comparison with GNN variants using dummy nodes.** In Zhang et al. (2022a); Liu et al. (2022), global dummy nodes are introduced and connected to all nodes for graph structure learning. Our global workspace method has advantages over dummy node-based methods: 1) Dummy nodes are compelled to receive information from all nodes, potentially causing over-squashing (Alon & Yahav, 2021). Global workspace, on the other hand, uses attention to selectively accept embeddings from a few crucial nodes. 2) In mini-batch training, dummy nodes fail to facilitate message-passing among out-of-batch nodes, whereas InterpGCN-GW effectively summarizes nodes from previous batches using memory. The performance comparison between dummy nodes and global workspace on CoraFull and AMiner-CS datasets is shown in Fig. 7.

**Improvement in fairness.** Graph structure-induced unfairness is worth studying, e.g., in social recommendation systems where most training samples from one community may lead to weaker generalization in another isolated communities. We explore structural unfairness by comparing the mean difference between the distribution of two node groups: those with direct connections to the

Table 7: Improvement in structure fairness. Smaller $\Delta_{SP}$, $\Delta_{EO}$ indicate higher fairness.

| Method | Cora | | Wiki-CS | |
|---|---|---|---|---|
| | $\Delta_{SP}$ | $\Delta_{EO}$ | $\Delta_{SP}$ | $\Delta_{EO}$ |
| GCN | 0.195 | 0.177 | 0.478 | 0.432 |
| InterpGCN-GW | 0.158 | 0.164 | 0.451 | 0.402 |

training set and those without paths to the training set. We use fairness metric *demographic parity* $\Delta_{DP}$ and *equal opportunity* $\Delta_{EO}$ following Liu et al. (2023) with results in Table 7,

$$\Delta_{\mathrm{DP}} = \tfrac{1}{K} \sum_{y \in Y} |\Pr(\hat{y}_i = y | i \in V_0) - \Pr(\hat{y}_i = y | i \in V_1)|,$$

$$\Delta_{\mathrm{EO}} = \tfrac{1}{K} \sum_{y \in Y} |\Pr(\hat{y}_i = y | y_i = y, i \in V_0) - \Pr(\hat{y}_i = y | y_i = y, i \in V_1)|.$$

Our results indicate that introducing a global workspace enables isolated/distant nodes to access training node information, reducing structural unfairness compared to GCN. Experiments are conducted on small graphs due to the computational impracticability of calculating the node-pair distance matrix on graphs with millions of nodes.

**Results on heterophilic graph datasets.** We have conducted experiments on heterophilic graph datasets, using official GitHub implementations with default hyperparameters for the baselines. The results are in Table 11. Observations on heterophilic graphs include: i) InterpGCN surpasses GCN on all three datasets, showing that enhancing distant node interplay via global workspace improves GCN's performance on heterophilic graphs. ii) As a plug-and-play module, our method combines well with other GNNs designed for heterophily, achieving improvements in these models. iii) Our method's performance is reliant on the base feature extractor. For instance, on the Squirrel dataset, where GCN actually performs better than other heterophilic GNNs, our GCN-based implementation also outperforms implementations based on other models. Detailed discussions on heterophilic graphs can be seen in Appendix G.3.

## 5 CONCLUSION AND FURTHER DISCUSSION

We derive a PAC-Bayes bound for message-passing GNNs in semi-supervised transductive learning. The bound associates the generalization error with the interplay between test and training nodes. We then propose the Graph Global Workspace method to enhance the interplay among distant nodes and out-of-batch nodes. Experimental results show the superiority on both small and large graphs.

**Limitations.** Based on our theoretical results, there could be more alternative techniques to enhance the generalization ability beyond our current technique. Also our work only covers the massage-passing scheme and there could emerge other forms of GNNs. **Broader Impact.** The proposed $L$-hop interplay can also be integrated into more frameworks for analyzing generalization bounds. Our theory may inspire more works on novel design for GNNs for both accuracy and fairness.

## 6 ACKNOWLEDGEMENTS

This work has been partially supported by the National Key R&D Program of China No. 2020YFB1710900 and No. 2020YFB1806700, NSFC Grants 61932014 and 62232011.

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
