# A    RELATED WORKS

**Graph Neural networks** We detail the message passing graph neural networks (Kipf & Welling, 2017; Veličković et al., 2018; Hamilton et al., 2017) for semi-supervised node classification. Graph neural networks innovates on capturing the dependency among nodes via message propagation. GNNs repeatedly aggregate the embedding of neighbors and combine the aggregated information with the original embedding. Generally, GNNs are categorized into spatial-domain and spectral-domain approaches. Based on the spectral graph theory, Bruna et al. (2014) first defines the graph convolution in the spectral domain through the eigen-decomposition of the graph Laplacian, defectively causing high computational cost. Graph Convolution Network (GCN) (Kipf & Welling, 2017) is utilized for our PAC-Bayes bounds analysis and feature extractor, which utilizes the 1-st approximation of the Chebyshev expansion to simplify the calculation. The $k$-th layer of GCN can be formally defined as:

$$\mathbf{H}_k = \sigma_k(\tilde{L}\mathbf{H}_{k-1}\mathbf{W}_k), \tag{12}$$

where $\sigma_k(\cdot)$ is the non-linear activation function, e.g., ReLU activation function, $W_k$ is the trainable weight matrix of the $k$-th layer, $H_k$ are node embeddings of the $k$-th layer and $H_0 = X$. The graph Laplacian is defined as $\tilde{L} = D^{\frac{1}{2}}\tilde{\mathbf{A}}D^{\frac{1}{2}}$, $\tilde{\mathbf{A}} = A + I$, where $D$ is the degree matrix $D_{ii} = \sum_j \tilde{\mathbf{A}}_{ij}$. Spatial-based approaches follow a message passing scheme (Abu-El-Haija et al., 2019), where each node collects the information from its neighbors iteratively. GraphSAGE (Hamilton et al., 2017) aggregates the information from randomly sampled neighborhoods to scale to large graphs. GAT (Veličković et al., 2018) introduces the attention mechanism to assign scores for each node pair. GIN (Xu et al., 2019) generalizes the Weisfeiler-Lehman test and reaches the most expressive power among GNNs. Ma et al. (2020); Khasahmadi et al. (2020) adds the memory layer on graph neural networks to model the long-range dependency. However, these memory-based GNNs do not consider the use of attention mechanisms to filter important nodes, while also neglecting the effect of positional encoding in node coordination, as demonstrated in our ablation studies.

**Theoretical Study on GNN's Generalization Bounds.** *Generalization* refers to the difference in model performance between the training and test sets. The first GNN generalization bound (Scarselli et al., 2018) directly links the generalization error with the VC dimension of GNN class. Since all subsequent GNN generalization bounds involve model size, training set size etc., we will only discuss parameters unique to graphs in the following. Du et al. (2019a) introduce Graph Neural Tangent Kernel, i.e., infinitely wide GNNs and establishes the generalization bound for graph classification. Oono & Suzuki (2020); Garg et al. (2020); Esser et al. (2021) derive bounds using Rademacher complexity. Liao et al. (2021) applies PAC-Bayesian analysis to message passing GNNs, revealing dependency on maximum node degree. Ma et al. (2021) analyze the transductive non-i.i.d. node classification task, same as our setting, and finds that the difference of aggregated features between training and test node affects generalization bounds. Table 1 thoroughly compares GNN generalization bounds, considering settings, frameworks, and graph-specific information affecting the bounds. Existing researches overlook the influence of graph topology on GNN generalization bounds.

**PAC-Bayesian** The PAC-Bayesian framework are Probably Approximately Correct (PAC) performance bounds for Bayesian learning algorithms. Initially introduced by McAllester (1998; 1999), the PAC-Bayesian framework has been further developed in subsequent works (Seeger, 2002; McAllester, 2003a;b; Maurer, 2004). A PAC bound (Valiant, 1984) represents the upper bound $\epsilon$ for the error-rate of the algorithm with high probability (*probably*). The algorithm is *approximately correct* when this bound is small. The result is referred to as PAC-Bayesian bounds when PAC bounds are applied to (generalized) Bayesian learning. Recently, the PAC-Bayesian has regained interest for deriving generalization bounds for neural networks. Neyshabur et al. (2017) and Dziugaite & Roy (2017) prove theoretical generalization bounds for deep neural networks based on results in McAllester (1998). leveraging the perturbation analysis from Neyshabur et al. (2017), Liao et al. (2021) derive a generalization bound for graph neural network on graph classification tasks, which is related to the maximum node degree and the spectral norm of the weights. Ma et al. (2021) obtain PAC-Bayesian generalization bounds for GNN in terms of the discrepancy between node features in the training set and the testing set. However, current PAC-Bayesian Analysis on GNNs overlook the the bias induced by the graph structure.

**Methods for Boosting GNNs' Generalization Ability.** Enhancing the interplay between distant nodes can boost GNNs' generalization ability but is challenging due to the over-smoothing. Methods for enhancing long-range interplay between nodes include three categories: 1) *Alleviating over-*

*smoothing for deep GNNs.* Researchers employ residual mechanisms (Li et al., 2019; Xu et al., 2018; Liu et al., 2021), edge dropping (Rong et al., 2020; Hasanzadeh et al., 2020), and normalization Zhao & Akoglu (2020); Bodnar et al. (2022) techniques to deepen GNNs. To address the neighbor explosion problem, Wu et al. (2019); Zeng et al. (2021) decouple the message passing and feature transformation procedures. 2) *Implicit connection.* Zhang et al. (2022a); Liu et al. (2022) introduce global dummy nodes connected to all nodes on the original graph, enabling long-range nodes to interact via dummy nodes. Graph transformers (Veličković et al., 2018; Wu et al., 2022; 2023; Ying et al., 2021) employ attention mechanisms to create virtual connections between arbitrary node pairs, allowing each node to aggregate information from all nodes. Alon & Yahav (2021) propose a rewiring method based on Balanced Forman curvature to solve the over-squashing problem. 3) *Historical embedding for large-scale graph.* As a mainstream method for scalable GNNs (Hamilton et al., 2017; Chiang et al., 2019; Zeng et al., 2020), mini-batch training based on subgraph sampling faces generalization reduction due to the absence of message passing between neighbor nodes across different batches. GNNAutoscale (Fey et al., 2021) stores historical node embeddings on the CPU and transfers 1-hop neighbor embeddings to the GPU for training of each epoch.

**Key-value attention and graph transformer.** In recent years, transformers (Vaswani et al., 2017) have shown superior performance across various field including computer vision (Dosovitskiy et al., 2020), natural language processing (Devlin et al., 2019) and graph (Kreuzer et al., 2021). As the backbone of transformer, key-value attention encodes relationships among input entities. selectively focusing on them. Key-value attention calculate the similarity between each query vector and corresponding key vector as the current state of the model. The obtained similarity score can then be used to measure the importance of the input value, and to guide the updating process for the global workspace in this paper. Specifically, the attention mechanism first projects an input tensor $\mathbf{X}$ into *queries* $\mathbf{Q}$, *keys* $\mathbf{K}$, and *values* $\mathbf{V}$ with different linear layers: $\mathbf{Q} = \mathbf{XW}_Q, \mathbf{K} = \mathbf{XW}_K, \mathbf{V} = \mathbf{XW}_V$. Apply the Softmax on query-key products to establish a probability distribution, subsequently generating output through the weighted summation of values based on similarity scores: $\mathbf{Z} := \text{Softmax}(\frac{\mathbf{QK}^\top}{\sqrt{d}})\mathbf{V}$. Given the success of transformers, scholars have extended it to the graph domain, especially for over-smoothing (Chen et al., 2020a) and over-squashing (Alon & Yahav, 2021) in Graph Neural Networks by modeling the long-range nodes dependency with attention. GraphTrans (Wu et al., 2021) applies a transformer module on top of a standard GNN layer to compute the pairwise node correlations in a position-agnostic way. GT (Dwivedi & Bresson, 2020) utilize the Laplacian eigen-vectors as the positional encoding and generalize the Transformer on graphs. Due to the quadratically computational complexity of transformer, most existing graph transformers are designed for small graph, which are hard to be applied on large-scale graph dataset with millions of nodes (Hu et al., 2020). Recent attempts to scale up the graph transformer to large graphs lie in sampling strategies. NAGphormer (Chen et al., 2023) aggregates multi-hop neighborhood features for each node as sequences of token. By taking each node as a sequence, the NAGphormer can be trained in a mini-batch method. NodeFormer (Wu et al., 2022) propose a kernelized Gumbel-Softmax operator which fuse random feature map and approximated sampling strategy to distil latent graph structures. Present researches try to employ transformer architectures to resolve the long-range node dependency challenges. However, these methods, predicted on pairwise attention, face difficulties when scaling to large graphs. Some approaches utilize sampling techniques for model training, yet they lack global consistency.

# B   PROOFS IN SECTION 2

We first introduce following lemma which is commonly used in PAC-Bayesian.

**Lemma 2 (Change of measure inequality (Germain et al., 2015))** *For any hypothesis class $\mathcal{H}$, for any distributions $P$ and $Q$ on $\mathcal{H}$, and for any measurable function $\phi : \mathcal{H} \to \mathbb{R}$*

$$\mathbb{E}_{f \sim Q}[\phi(f)] \leq \mathcal{D}_{KL}(Q\|P) + \ln\left(\mathbb{E}_{f \sim P}[\phi(f)]\right)$$

**Lemma 3 (Markov's inequality )** *Let $X$ be a non-negative random variable and $\alpha > 0$, we have:*

$$Pr(X > a\mathbb{E}[X]) < 1/a$$

**Lemma 4 (Jensen's Inequality (Jensen, 1906))** *If $f$ is a convex function, and $\mathbb{E}[f(X)]$ and $f(\mathbb{E}[X])$ are finite, then*

$$f(\mathbb{E}[X]) \leq \mathbb{E}[f(X)]$$

**Lemma 5 (Hoeffding inequality (Hoeffding, 1994))** *Let* $X_1, X_2, \cdots, X_N$ *be i.i.d. random variables bounded in [0,1]. Then for all* $\epsilon > 0$,

$$\Pr\left[\left|\frac{1}{N}\sum_{n=1}^{N} Z_n - \mathbb{E}\left[Z_n\right]\right| > \epsilon\right] \le 2e^{\left(-2N\epsilon^2\right)} \tag{13}$$

**Lemma 6 (Perturbation Bound of GCN)** *(Adopted from Lemma 2 in Neyshabur et al. (2017)) For any* $B, L > 0$, *let* $f_{\mathbf{W}}$ *be a* $L$-*layer graph convolution network with ReLU activation. Then for any* $\mathbf{w}$ *and* $x$, *and any perturbation* $\mathbf{u} = vec(\{\mathbf{U}_i\}_{i=1}^{L})$ *such that* $\|\mathbf{U}_i\| \le \frac{1}{L}\|\mathbf{W}_i\|_2$, *the change in the output of the network can be bounded as follows:*

$$|f_{\mathbf{u}+\mathbf{w}}(\mathbf{A}, \mathbf{X})[i,:] - f_{\mathbf{w}}(\mathbf{A}, \mathbf{X})[i,:]|_2 \le eB(\prod_{i=1}^{L}\|\mathbf{W}_i\|_2)\sum_{i=1}^{L}\frac{\|\mathbf{U}_i\|_2}{\|\mathbf{W}_i\|_2}. \tag{14}$$

## B.1 PROOF OF LEMMA 1

**Lemma 1 (PAC-Bayesian bound for transductive learning)** *Given a full set* $V$ *with* $N$ *examples where each example* $(x, y)$ *is i.i.d drawn from distribution* $D$, *for any prior distribution* $P$ *on the hypothesis space of classifier* $\mathcal{H}$, *for any* $\delta \in (0, 1]$, *with probability at least* $1 - \delta$, *we have,*

$$r_{U,\gamma}(Q) \le R_{S,\gamma}(Q) + \sqrt{\frac{\mathcal{D}_{KL}(Q\|P) + \ln\frac{2m+2}{\delta} + D(P)}{2m}} \tag{15}$$

*Proof.* Consider the random variable $\Delta\left(r_{U,0}(Q), R_{S,\gamma}(Q)\right) := \left(r_{U,\gamma}(Q) - r_{S,\gamma}(Q)\right)^2$. We aims to upper bound $\Delta$ with high probability. Since the $\Delta$ is convex, with Jensen's inequality,

$$\Delta\left(r_{U,\gamma}(Q), R_{S,\gamma}(Q)\right) \le \mathop{\mathbb{E}}_{f\sim Q}[(r_{U,\gamma}(f) - R_{S,\gamma}(f))^2]. \tag{16}$$

To transform the expectation on $Q$ into the expectation on $P$ which is independent on $S$, set $\phi(f) = 2(m-1)(r_{U,\gamma}(f) - R_{S,\gamma}(f))^2$ in Lemma 2, we have:

$$\mathop{\mathbb{E}}_{f\sim Q}\left[(r_{U,\gamma}(f) - R_{S,\gamma}(f))^2\right] \le \frac{1}{2(m-1)}\left(\mathcal{D}_{KL}[Q\|P] + \ln\mathop{\mathbb{E}}_{f\sim P}\left[e^{2(m-1)(r_{U,\gamma}(f) - R_{S,\gamma}(f))^2}\right]\right). \tag{17}$$

Applying Markov's inequality to $\mathbb{E}_{f\sim P}\left[e^{2(m-1)(r_{U,\gamma}(f) - R_{S,\gamma}(f))^2}\right]$, we have for any $\delta > 0$, with probability at least $1 - \delta$,

$$\ln\mathop{\mathbb{E}}_{f\sim P}\left[e^{2(m-1)(r_{U,\gamma}(f) - R_{S,\gamma}(f))^2}\right]$$
$$\le \ln\frac{1}{\delta}\mathop{\mathbb{E}}_{y\sim D}\left[\mathop{\mathbb{E}}_{f\sim P}\left[e^{2(m-1)(r_{U,\gamma}(f) - R_{S,\gamma}(f))^2}\right]\right]$$
$$= \ln\frac{1}{\delta}\mathop{\mathbb{E}}_{f\sim P}\mathop{\mathbb{E}}_{y\sim D}\left[e^{2(m-1)(r_{U,\gamma}(f) - R_{S,\gamma}(f))^2}\right] \tag{18}$$

For any $f \in \mathcal{H}$, we have,

$$\mathop{\mathbb{E}}_{y\sim D}\left[e^{2(m-1)(r_{U,\gamma}(f) - R_{S,\gamma}(f))^2}\right]$$
$$= \mathop{\mathbb{E}}_{y\sim D}\left[e^{2(m-1)(r_{U,\gamma}(f) + r_{S,\gamma}(f) - r_{S,\gamma}(f) - R_{S,\gamma}(f))^2}\right]$$
$$\le \mathop{\mathbb{E}}_{y\sim D}\left[e^{2(m-1)(r_{U,\gamma}(f) - r_{S,\gamma}(f))^2 + 2(m-1)(r_{S,\gamma}(f) - R_{S,\gamma}(f))^2}\right]$$
$$= e^{2(m-1)(r_{U,\gamma}(f) - r_{S,\gamma}(f))^2}\mathop{\mathbb{E}}_{y\sim D}\left[e^{2(m-1)(r_{S,\gamma}(f) - R_{S,\gamma}(f))^2}\right] \tag{19}$$

Let $X$ be a non-negative random variable, then we have,

$$
\begin{aligned}
\mathbb{E}[X] &= \int_0^\infty X \mathrm{Pr}(X) dX \\
&= \int_0^\infty \int_0^X 1 d\epsilon \mathrm{Pr}(X) dX \\
&= \int_0^\infty \int_0^X \mathrm{Pr}(X) d\epsilon dX \\
&= \int_0^\infty \int_\epsilon^\infty \mathrm{Pr}(X) dX d\epsilon \\
&= \int_0^\infty \mathrm{Pr}(X \geq \epsilon) d\epsilon
\end{aligned}
$$

Let $X = |r_{S,\gamma}(f) - R_{S,\gamma}(f)|$, we have

$$
\mathbb{E}_{y \sim D}[|r_{S,\gamma}(f) - R_{S,\gamma}(f)|] = \int_0^\infty \mathrm{Pr}(|r_{S,\gamma}(f) - R_{S,\gamma}(f)| > \epsilon) d\epsilon \tag{20}
$$

By Hoeffding inequality in Lemma 5, we have,

$$
\mathrm{Pr}\left(|r_{S,\gamma}(f) - R_{S,\gamma}(f)| \geq \epsilon\right) \leq 2e^{-2m\epsilon^2} \tag{21}
$$

Substitute Eq. 20 and Eq. 21 into Eq. 19 we have,

$$
\begin{aligned}
&\mathbb{E}_{y \sim D}\left[e^{2(m-1)(r_{S,\gamma}(f) - R_{S,\gamma}(f))^2}\right] \\
=\ & \int_0^\infty \mathrm{Pr}\left((r_{S,\gamma}(f) - R_{S,\gamma}(f))^2 \geq \frac{\ln \epsilon}{2(m-1)}\right) d\epsilon \\
\leq\ & \int_0^\infty \mathrm{Pr}\left((r_{S,\gamma}(f) - R_{S,\gamma}(f)) \geq \sqrt{\frac{\ln \epsilon}{2(m-1)}}\right) d\epsilon + \int_0^\infty \mathrm{Pr}\left(r_{S,\gamma}(f) - R_{S,\gamma}(f) \leq -\sqrt{\frac{\ln \epsilon}{2(m-1)}}\right) d\epsilon
\end{aligned}
$$

$$
\begin{aligned}
&\int_0^\infty \mathrm{Pr}\left((r_{S,\gamma}(f) - R_{S,\gamma}(f)) \geq \sqrt{\frac{\ln \epsilon}{2(m-1)}}\right) d\epsilon \\
=\ & \int_0^1 \mathrm{Pr}\left((r_{S,\gamma}(f) - R_{S,\gamma}(f)) \geq \sqrt{\frac{\ln \epsilon}{2(m-1)}}\right) d\epsilon + \int_1^\infty \mathrm{Pr}\left((r_{S,\gamma}(f) - R_{S,\gamma}(f)) \geq \sqrt{\frac{\ln \epsilon}{2(m-1)}}\right) d\epsilon \\
\leq\ & 1 + \int_1^\infty e^{-2m \frac{\ln \epsilon}{2(m-1)}} d\epsilon \\
=\ & m
\end{aligned}
$$

$$
\int_0^\infty \mathrm{Pr}\left((r_{S,\gamma}(f) - R_{S,\gamma}(f)) \leq -\sqrt{\frac{\ln \epsilon}{2(m-1)}}\right) d\epsilon \leq m \tag{22}
$$

$$
\mathbb{E}_{y \sim D}\left[e^{2(m-1)(r_{S,\gamma}(f) - R_{S,\gamma}(f))^2}\right] \leq 2m \tag{23}
$$

Substitute Eq. 23 and Eq. 19 into Eq. 18, we have,

$$
\begin{aligned}
&\ln \frac{1}{\delta} \mathbb{E}_{f \sim P} \mathbb{E}_{y \sim D}\left[e^{2(m-1)(r_{U,\gamma}(f) - R_{S,\gamma}(f))}\right] \\
\leq\ & \ln \mathbb{E}_{f \sim P}\left(\frac{2m}{\delta} \cdot \mathbb{E}_{y \sim D}\left[e^{2(m-1)(r_{U,\gamma}(f) - r_{S,\gamma}(f))}\right]\right)
\end{aligned} \tag{24}
$$

Then we finally have,

$$r_{U,0}(Q) - R_{S,\gamma}(Q) = \mathop{\mathbb{E}}_{f \sim Q}[(r_{U,\gamma}(f) - R_{S,\gamma}(f))]$$

$$\leq \sqrt{\frac{\mathcal{D}_{\mathrm{KL}}[Q\|P] + \ln \mathop{\mathbb{E}}_{f \sim P}\left[e^{2(m-1)(r_{U,\gamma}(f) - R_{S,\gamma}(f))^2}\right]}{2(m-1)}}$$

$$\leq \sqrt{\frac{\mathcal{D}_{\mathrm{KL}}[Q\|P] + \ln \mathop{\mathbb{E}}_{f \sim P}\left(\frac{2m}{\delta} \cdot \mathop{\mathbb{E}}_{f \sim P}\left[e^{2m(r_{U,\gamma}(f) - r_{S,\gamma}(f))^2}\right]\right)}{2(m-1)}}$$

$$= \sqrt{\frac{\mathcal{D}_{\mathrm{KL}}[Q\|P] + D(P) + \ln \frac{2m}{\delta}}{2(m-1)}} \tag{25}$$

where $D(P) = \ln \mathop{\mathbb{E}}_{f \sim P}\left[e^{2(m-1)(r_{U,\gamma}(f) - r_{S,\gamma}(f))^2}\right]$.  □

## B.2 Proof of Theorem 1

We first introduce the following lemma adpoted from Lemma 1 in Neyshabur et al. (2017).

**Lemma 7** *Let $f_{\mathbf{w}}$ be any classifier with parameters $\mathbf{w}$ (not necessarily a graph neural network), For any prior distribution $P$, the posterior $Q$ is built by adding a random perturbation $\mathbf{u}$ to $\mathbf{w}$ s.t., $\Pr(\max_{i \in V} |f_{\mathbf{w}+\mathbf{u}}(\mathbf{A}, \mathbf{X})[i,:] - f_{\mathbf{w}}(\mathbf{A}, \mathbf{X})[i,:]|_\infty \leq \frac{\gamma}{4}) > \frac{1}{2}$. For any $\gamma, \delta > 0$, with probability at least $1 - \delta$ of the choice of training set, we have:*

$$r_{U,0}(f_{\mathbf{w}}) \leq R_{S,\gamma}(f_{\mathbf{w}}) + \sqrt{\frac{2\mathcal{D}_{KL}(Q\|P) + \ln \frac{8m}{\delta} + D(P)}{2(m-1)}}$$

*Proof.* Let $\tilde{\mathbf{w}} = \mathbf{w} + \mathbf{u}$. Let $\mathcal{P}$ be the set of perturbations s.t.:

$$\mathcal{P} = \left\{\tilde{\mathbf{w}} \,\middle|\, \max_{i \in S \cup U} |f_{\mathbf{w}'}(\mathbf{A}, \mathbf{X})[i,:] - f_{\mathbf{w}}(\mathbf{A}, \mathbf{X})[i,:]|_\infty < \frac{\gamma}{4}\right\}$$

We construct a new posterior $\tilde{Q}$ as

$$\tilde{Q}(\tilde{\mathbf{w}}) = \begin{cases} \frac{1}{Z}Q(\tilde{\mathbf{w}}), & \tilde{\mathbf{w}} \in \mathcal{P} \\ 0, & \text{otherwise} \end{cases} \tag{26}$$

where $Z = \Pr_{\tilde{\mathbf{w}} \sim Q}(\tilde{\mathbf{w}} \in \mathcal{P}) \geq \frac{1}{2}$ is a normalizing constant. By definition of $\tilde{Q}$, for any $\tilde{\mathbf{w}} \sim \tilde{Q}$ and any nodes $i$, we have

$$\max_{k_1, k_2 \in [K]} ||f_{\tilde{\mathbf{w}}}(\mathbf{A}, \mathbf{X})[i, k_1] - f_{\tilde{\mathbf{w}}}(\mathbf{A}, \mathbf{X})[i, k_2]| - |f_{\mathbf{w}}(\mathbf{A}, \mathbf{X})[i, k_1] - f_{\mathbf{w}}(\mathbf{A}, \mathbf{X})[i, k_2]|| \leq \frac{\gamma}{2} \tag{27}$$

Let $\mathcal{M}_{\mathbf{w}}(x_i, y_i) = f_{\mathbf{w}}(\mathbf{A}, \mathbf{X})[i, y_i] - \max_{k \neq y_i} f_{\mathbf{w}}(\mathbf{A}, \mathbf{X})[i, k]$ and $\mathcal{M}_{\tilde{\mathbf{w}}}(x_i, y_i) = f_{\tilde{\mathbf{w}}}(\mathbf{A}, \mathbf{X})[i, y_i] - \max_{k \neq y_i} f_{\mathbf{w}}(\mathbf{A}, \mathbf{X})[i, k]$. Recall that

$$\Phi^0(\mathcal{M}_{\mathbf{w}}(x_i, y_i)) = \begin{cases} 0 & 0 \leq \mathcal{M}_{\mathbf{w}}(x_i, y_i) \\ 1 & \mathcal{M}_{\mathbf{w}}(x_i, y_i) \leq 0 \end{cases}$$

$$\Phi^{\gamma/2}(\mathcal{M}_{\tilde{\mathbf{w}}}(x_i, y_i)) = \begin{cases} 0 & \gamma \leq \mathcal{M}_{\tilde{\mathbf{w}}}(x_i, y_i) \\ 1 - 2\mathcal{M}_{\tilde{\mathbf{w}}}(x_i, y_i)/\gamma & 0 \leq \mathcal{M}_{\tilde{\mathbf{w}}}(x_i, y_i) \leq \gamma \\ 1 & \mathcal{M}_{\tilde{\mathbf{w}}}(x_i, y_i) \leq 0 \end{cases} \tag{28}$$

From Eq. 27, we have,

$$|\mathcal{M}_{\mathbf{w}}(x_i, y_i) - \mathcal{M}_{\tilde{\mathbf{w}}}(x_i, y_i)| \leq \gamma/2 \tag{29}$$

If $\mathcal{M}_{\mathbf{w}}(x_i, y_i) > \mathcal{M}_{\tilde{\mathbf{w}}}(x_i, y_i)$, we have,

$$\mathcal{M}_{\mathbf{w}}(x_i, y_i) \leq \mathcal{M}_{\tilde{\mathbf{w}}}(x_i, y_i) + \frac{\gamma}{2}$$

Thus we surely have,

$$\Phi^0(\mathcal{M}_{\mathbf{w}}(x_i, y_i)) \le \Phi^{\gamma/2}(\mathcal{M}_{\tilde{\mathbf{w}}}(x_i, y_i)) \tag{30}$$

If $\mathcal{M}_{\mathbf{w}}(x_i, y_i) < \mathcal{M}_{\tilde{\mathbf{w}}}(x_i, y_i)$, we directly have

$$\Phi^0(\mathcal{M}_{\mathbf{w}}(x_i, y_i)) \le \Phi^{\gamma/2}(\mathcal{M}_{\tilde{\mathbf{w}}}(x_i, y_i)) \tag{31}$$

Eq. 30 and Eq. 31 indicates that

$$r_{U,0}(f_{\mathbf{w}}) \le r_{U,\gamma/2}(f_{\tilde{\mathbf{w}}}) \tag{32}$$

Similarly, we have

$$R_{S,\gamma/2}(f_{\tilde{\mathbf{w}}}) \le R_{S,\gamma}(f_{\mathbf{w}}) \tag{33}$$

Then, with probability $1 - \delta$ over the sample of label, we have:

$$
\begin{aligned}
& r_{U,0}(f_{\mathbf{w}}) \\
\le \;& \mathop{\mathbb{E}}_{\tilde{\mathbf{w}} \sim \tilde{Q}} [r_{U,\gamma/2}(f_{\tilde{\mathbf{w}}})] \\
\le \;& \mathop{\mathbb{E}}_{\tilde{\mathbf{w}} \sim \tilde{Q}} [R_{S,\gamma/2}(f_{\tilde{\mathbf{w}}})] + \sqrt{\frac{\mathcal{D}_{KL}(\tilde{Q}\|P) + \ln \frac{2m}{\delta} + D(P)}{2(m-1)}} \\
\le \;& R_{S,\gamma}(f_{\mathbf{w}}) + \sqrt{\frac{\mathcal{D}_{KL}(\tilde{Q}\|P) + \ln \frac{2m}{\delta} + D(P)}{2(m-1)}}
\end{aligned}
\tag{34}
$$

Denote $\mathcal{P}^c$ as the complement of $\mathcal{P}$. Define $H(Z) := -Z \ln Z - (1-Z)\ln(1-Z)$ as the binary entropy function,

$$
\begin{aligned}
& \mathcal{D}_{KL}(Q\|P) \\
= \;& \int_{\tilde{\mathbf{w}} \in \mathcal{P}} Q \ln \frac{Q}{P} d\tilde{\mathbf{w}} + \int_{\tilde{\mathbf{w}} \in \mathcal{P}^c} Q \ln \frac{Q}{P} d\tilde{\mathbf{w}} \\
= \;& Z \mathcal{D}_{KL}(\tilde{Q}\|P) + (1-Z)\mathcal{D}_{KL}(\tilde{Q}^c\|P) - H(Z),
\end{aligned}
$$

where $\tilde{Q}^c$ is the normalized density of $Q$ restricted to $\mathcal{P}^c$. Since KL divergence is always positive, so we have,

$$\mathcal{D}_{KL}(\tilde{Q}\|P) = \frac{1}{Z}[\mathcal{D}_{KL}(Q\|P) + H(Z) - (1-Z)\mathcal{D}_{KL}(Q^c\|P)] \le 2(\mathcal{D}_{KL}(Q\|P) + \ln 2) \tag{35}$$

Thus,

$$r_{U,0}(f_{\mathbf{w}}) \le R_{S,\gamma}(f_{\mathbf{w}}) + \sqrt{\frac{2\mathcal{D}_{KL}(Q\|P) + \ln \frac{8m}{\delta} + D(P)}{2(m-1)}}. \tag{36}$$

$\square$

**Theorem 1 (PAC-Bayes bounds with $L$-hop interplay for transductive GCN)** Let $f \in \mathcal{H}$ be an $L$-layer GCN with parameters $\{\mathbf{W}_i\}_{i=1}^L$, for any $B > 0$, $L, h \ge 1$, and any $\delta, \gamma > 0$, with probability at least $1 - \delta$ over a training set $S$ of size $m$ we have,

$$r_{U,0}(f) \le R_{S,\gamma}(f) + O\left(\sqrt{\frac{B^2 L^2 h \ln(4Lh)\Pi\Sigma + \ln \frac{Lm}{\delta}}{\gamma^2 m}} + K^2 B^2 (1 - \frac{2\mathcal{I}_L}{d_{max}})^2 \Pi\right)$$

where $\Pi = \prod_{l=1}^L \|\mathbf{W}_i\|_F^2$ and $\Sigma = \sum_{l=1}^L \frac{\|\mathbf{W}_l\|_F^2}{\|\mathbf{W}_l\|_2^2}$ are product and sum of spectral norm of weights, $d_{max}$ is the maximum node degree.

*Proof.* In transductive learning, we can assume that the randomness of selecting the training and test sets is attributed to the sample of node labels with out loss of generality. For simplicity, we denote $f(\mathbf{A}, \mathbf{X})[j, :]$ as $f_j$ and denote true label-generating probability $\Pr(y_j = k|\mathbf{X}, G)$ as $\eta_k(j)$. We also denote the loss for one sample as $\mathcal{L}^\gamma(f_j, y_j) = \Phi^\gamma(\mathcal{M}_f(j, y_j)) = \Phi^\gamma(f_j[y_j] - \max_{k \ne y_j} h_j[k])$

where $\Phi^\gamma$ is defined in Eq. 2. We first bound the term $\ln \mathbb{E}_{f \sim P}\left[e^{2(m-1)(r_{U,\gamma}(f)-r_{S,\gamma}(f))^2}\right]$ which is the expected loss between training and test sets. For any $f \sim P$, we have:

$$r_{U,\gamma}(f) - r_{S,\gamma}(f)$$

$$= \mathbb{E}_{y \sim D}\left[\frac{1}{N-m}\sum_{j \in U}\mathcal{L}^\gamma(f_j, y_j)\right] - \mathbb{E}_{y \sim D}\left[\frac{1}{m}\sum_{i \in S}\mathcal{L}^\gamma(f_i, y_i)\right]$$

$$= \frac{1}{N-m}\sum_{j \in U}\sum_{k=1}^{K}(\eta_k(j)\mathcal{L}^\gamma(f_j, k)) - \frac{1}{m}\sum_{i \in S}\sum_{k=1}^{K}(\eta_k(i)\mathcal{L}^\gamma(f_i, k))$$

$$= \frac{1}{(N-m) \cdot m}\sum_{j \in U}\sum_{i \in S}\sum_{k=1}^{K}(\eta_k(j)\mathcal{L}^\gamma(f_j, k)) - \frac{1}{m \cdot (N-m)}\sum_{i \in S}\sum_{j \in U}\sum_{k=1}^{K}(\eta_k(i)\mathcal{L}^\gamma(f_i, k))$$

$$= \frac{1}{(N-m) \cdot m}\sum_{j \in U}\sum_{i \in S}\sum_{k=1}^{K}(\eta_k(j)\mathcal{L}^\gamma(f_j, k) - \eta_k(i)\mathcal{L}^\gamma(f_i, k))$$

$$= \frac{1}{(N-m) \cdot m}\sum_{j \in U}\sum_{i \in S}\sum_{k=1}^{K}(\eta_k(j)\mathcal{L}^\gamma(f_j, k) + \eta_k(j)\mathcal{L}^\gamma(f_i, k) - \eta_k(j)\mathcal{L}^\gamma(f_i, k) - \eta_k(i)\mathcal{L}^\gamma(f_i, k))$$

$$= \frac{1}{m \cdot (N-m)}\sum_{j \in U}\sum_{i \in S}\sum_{k=1}^{K}(\eta_k(j)(\mathcal{L}^\gamma(f_j, k) - \mathcal{L}^\gamma(f_i, k)) + (\eta_k(j) - \eta_k(i))\mathcal{L}^\gamma(f_i, k)) \quad (37)$$

Since $\eta_k(j)$ and $\mathcal{L}(h_i, k)$ are both bounded by 1:

$$r_{U,\gamma}(f) - r_{S,\gamma}(f)$$

$$\leq \frac{1}{m \cdot (N-m)}\sum_{j \in U}\sum_{i \in S}\sum_{k=1}^{K}(1 \cdot (\mathcal{L}^\gamma(f_j, k) - \mathcal{L}^\gamma(f_i, k)) + (\eta_k(j) - \eta_k(i)) \cdot 1) \quad (38)$$

And we have:

$$\frac{1}{m \cdot (N-m)}\sum_{j \in U}\sum_{i \in S}\sum_{k=1}^{K}((\eta_k(j) - \eta_k(i)))$$

$$= \frac{1}{m \cdot (N-m)}\sum_{j \in U}\sum_{i \in S}\left(\sum_{k=1}^{K}\eta_k(j) - \sum_{k=1}^{K}\eta_k(i)\right)$$

$$= 0 \quad (39)$$

Chose coordinate $k' \neq k$ so that $\mathcal{M}(f_i, k) = f_i[k] - f_i[k']$. Then

$$\mathcal{L}^\gamma(f_j, k) - \mathcal{L}^\gamma(f_i, k)$$

$$= \Phi^\gamma(\mathcal{M}(f_j, k)) - \Phi^\gamma(\mathcal{M}(f_i, k))$$

$$\leq \frac{1}{\gamma}|\mathcal{M}_f(j, k) - \mathcal{M}_f(i, k)| \quad \text{(Lipschitz property of ramp loss)}$$

$$= \frac{1}{\gamma}|\left(f_j[k] - \max_{l \neq k} f_j[l]\right) - (f_i[k] - f_i[k'])|$$

$$= \frac{1}{\gamma}|f_j[k] - f_i[k] + f_i[k'] + \min_{l \neq k}(-v_l)|$$

$$\leq \frac{1}{\gamma}|(f_j[k] - f_i[k]) + (f_i[k'] - f_j[k'])|$$

$$\leq \frac{2}{\gamma}\|f_j - f_i\|_\infty \leq \frac{2}{\gamma}\|f_j - f_i\|_2 = \frac{2}{\gamma}\|\mathbf{H}[i,:] - \mathbf{H}[j,:]\|_2 \quad (40)$$

where $\mathbf{H}$ is the output of GNNs' final layer. When consider the final layer of the GCN, we only need to aggregate the 1-hop neighbors of the nodes, we have:

$$\frac{1}{m \cdot (N-m)} \sum_{j \in U} \sum_{i \in S} \sum_{k=1}^{K} \left( \mathcal{L}^{\gamma}(f_j, k) - \mathcal{L}^{\gamma}(f_i, k) \right)$$

$$\leq \frac{2K}{\gamma \cdot m \cdot (N-m)} \sum_{j \in U} \sum_{i \in S} \|\mathbf{H}[i,:] - \mathbf{H}[j,:]\|_2$$

$$= \frac{2K}{\gamma \cdot m \cdot (N-m)} \sum_{j \in U} \sum_{i \in S} \|\sigma(\tilde{\mathbf{A}}\mathbf{H}'\mathbf{W}_L[j,:]) - \sigma(\tilde{\mathbf{A}}\mathbf{H}'\mathbf{W}_L[i,:])\|_2$$

$$\leq \frac{2K}{\gamma \cdot m \cdot (N-m)} \sum_{j \in U} \sum_{i \in S} \|\tilde{\mathbf{A}}\mathbf{H}'\mathbf{W}_L[j,:] - \tilde{\mathbf{A}}\mathbf{H}'\mathbf{W}_L[i,:]\|_2 \quad \text{(Lipschitz property of ReLU)}$$

$$= \frac{2K}{\gamma \cdot m \cdot (N-m)} \sum_{j \in U} \sum_{i \in S} \| \sum_{p \in \mathcal{N}_j^1} \tilde{\mathbf{A}}[j,p]\mathbf{H}'[p,:]\mathbf{W}_L - \sum_{q \in \mathcal{N}_i^1} \tilde{\mathbf{A}}[i,q]\mathbf{H}'[q,:]\mathbf{W}_L\|_2$$

where $\tilde{\mathbf{A}}[j,p]$ indicates the connectivity between node $j$ and $p$, $\mathcal{N}_j^1$ denotes the 1-hop neighbors of node $j$, and $\mathbf{H}'[p,:]$ represents the embeddings of node $p$ from previous layer. When considering the final layer of the GNN, for any test node $j \in U$, the update of its embedding aggregated embeddings of nodes including itself and its 1-hop neighbors. These 1-hop neighbors can be categorized into training and test nodes, i.e., $\mathcal{N}_j^1 = \mathcal{N}_j^1 \cap S + \mathcal{N}_j^1 \cap U$. We choose the adjacency matrix to be normalized by degree for simplicity. Given $\tilde{\mathbf{A}}[j,p] = \frac{1}{d_j}$ if $p \in \mathcal{N}_j^1$ and 0 otherwise, the above equation becomes:

$$\frac{1}{m \cdot (N-m)} \sum_{j \in U} \sum_{i \in S} \sum_{k=1}^{K} \left( \mathcal{L}^{\gamma/2}(f_j, k) - \mathcal{L}^{\gamma}(f_i, k) \right)$$

$$\leq \frac{2K}{\gamma \cdot m \cdot (N-m)} \sum_{j \in U} \sum_{i \in S} \|\frac{1}{d_j} \sum_{p \in \mathcal{N}_j^1} \mathbf{H}'[p,:]\mathbf{W}_L - \frac{1}{d_i} \sum_{q \in \mathcal{N}_i^1} \mathbf{H}'[q,:]\mathbf{W}_L\|_2$$

$$= \frac{2K}{\gamma \cdot m \cdot (N-m)} \sum_{j \in U} \sum_{i \in S} \left\| \frac{1}{d_j} \left( \mathbf{H}'[j] + \sum_{p \in \mathcal{N}_j^1 \cap U} \mathbf{H}'[p] + \sum_{p \in \mathcal{N}_j^1 \cap S} \mathbf{H}'[p] \right) \mathbf{W}_L \right.$$

$$\left. - \frac{1}{d_i} \left( \mathbf{H}'[i] + \sum_{q \in \mathcal{N}_i^1 \cap U} \mathbf{H}'[q] + \sum_{q \in \mathcal{N}_i^1 \cap S} \mathbf{H}'[q] \right) \mathbf{W}_L \right\|_2 \quad (41)$$

Only the nodes on the boundary can conduct message passing across training and test sets. *The message passing across training and test nodes can decrease the difference between embedding of training and test nodes.* We choose the coordinate of one training node $i^*$ and of one test node $j^*$ such that the difference between them is maximum, $i^*, j^* = \arg\max_{j^* \in U, i^* \in S} \|\mathbf{H}'[j^*,:] - \mathbf{H}'[i^*,:]\|_2$. We also denote $\tilde{\mathbf{H}}' := \|\mathbf{H}'[j^*,:] - \mathbf{H}'[i^*,:]\|_2$. Next, we will discuss based on condition of distance between $i$ and $j$.

①: $s(i, j) > 1$, then $i \notin \mathcal{N}_j^1 \cap S$ and $j \notin \mathcal{N}_i^1 \cap U$. Thus two nodes cannot exchange information.

$$
\begin{aligned}
&\|\mathbf{H}[j, :] - \mathbf{H}[i, :]\|_2 \\
=\ & \left\| \frac{1}{d_j} \left( \mathbf{H}'[j, :] + \sum_{p \in \mathcal{N}_j^1 \cap U} \mathbf{H}'[p, :] + \sum_{p \in \mathcal{N}_j^1 \cap S} \mathbf{H}'[p, :] \right) \mathbf{W}_L \right. \\
& \left. - \frac{1}{d_i} \left( \mathbf{H}'[i, :] + \sum_{q \in \mathcal{N}_i^1 \cap U} \mathbf{H}'[q, :] + \sum_{q \in \mathcal{N}_i^1 \cap S} \mathbf{H}'[q, :] \right) \mathbf{W}_L \right\|_2 \\
\leq\ & \left\| \frac{1}{d_j} \left( \mathbf{H}'[j^*, :] + \sum_{p \in \mathcal{N}_j^1 \cap U} \mathbf{H}'[j^*, :] + \sum_{p \in \mathcal{N}_j^1 \cap S} \mathbf{H}'[j^*, :] \right) \mathbf{W}_L \right. \\
& \left. - \frac{1}{d_i} \left( \mathbf{H}'[i^*, :] + \sum_{q \in \mathcal{N}_i^1 \cap U} \mathbf{H}'[i^*, :] + \sum_{q \in \mathcal{N}_i^1 \cap S} \mathbf{H}'[i^*, :] \right) \mathbf{W}_L \right\|_2 \\
\leq\ & \|\mathbf{H}'[j^*, :] - \mathbf{H}'[i^*, :]\|_2 \|\mathbf{W}_L\|_2
\end{aligned}
\tag{42}
$$

②: $s(i, j) = 1$, then $i \in \mathcal{N}_j^1 \cap S$ and $j \in \mathcal{N}_i^1 \cap U$. Thus two nodes can exchange information. Eq. 41 becomes:

$$
\begin{aligned}
&\|\mathbf{H}[j, :] - \mathbf{H}[i, :]\|_2 \\
=\ & \left\| \frac{1}{d_j} \left( \mathbf{H}'[j, :] + \sum_{p \in \mathcal{N}_j^1 \cap U} \mathbf{H}'[p, :] + \sum_{p \in \mathcal{N}_j^1 \cap S} \mathbf{H}'[p, :] \right) \mathbf{W}_L \right. \\
& \left. - \frac{1}{d_i} \left( \mathbf{H}'[i, :] + \sum_{q \in \mathcal{N}_i^1 \cap U} \mathbf{H}'[q, :] + \sum_{q \in \mathcal{N}_i^1 \cap S} \mathbf{H}'[q, :] \right) \mathbf{W}_L \right\|_2 \\
=\ & \left\| \frac{1}{d_j} \left( \mathbf{H}'[j, :] + \sum_{p \in \mathcal{N}_j^1 \cap U} \mathbf{H}'[j, :] + \sum_{p \in \mathcal{N}_j^1 \cap S/\{i\}} \mathbf{H}'[p, :] + \mathbf{H}[i, :] \right) \mathbf{W}_L \right. \\
& \left. - \frac{1}{d_i} \left( \mathbf{H}'[i, :] + \sum_{q \in \mathcal{N}_i^1 \cap U/\{j\}} \mathbf{H}'[q, :] + \sum_{q \in \mathcal{N}_i^1 \cap S} \mathbf{H}'[q, :] + \mathbf{H}'[j, :] \right) \mathbf{W}_L \right\|_2 \\
\leq\ & \left\| \frac{(\mathbf{H}'[j, :] + (d_j - 2)\mathbf{H}'[j^*, :] + \mathbf{H}'[i, :])}{d_j} \mathbf{W}_L - \frac{(\mathbf{H}'[i, :] + (d_i - 2)\mathbf{H}'[i^*, :] + \mathbf{H}'[j, :])}{d_i} \mathbf{W}_L \right\|_2 \\
=\ & \left\| (\mathbf{H}'[j^*, :] - \mathbf{H}'[i^*, :]) - \frac{2\mathbf{H}'[j^*, :] - \mathbf{H}'[i, :] - \mathbf{H}'[j, :]}{d_j} - \frac{\mathbf{H}'[j, :] + \mathbf{H}'[i, :] - 2\mathbf{H}'[i^*, :]}{d_i} \right\|_2 \|\mathbf{W}_L\|_2 \\
\leq\ & \left\| (\mathbf{H}'[j^*, :] - \mathbf{H}'[i^*, :]) - \frac{2\mathbf{H}'[j^*, :] - \mathbf{H}'[i, :] - \mathbf{H}'[j, :]}{d_{max}} - \frac{\mathbf{H}'[j, :] + \mathbf{H}'[i, :] - 2\mathbf{H}'[i^*, :]}{d_{max}} \right\|_2 \|\mathbf{W}_L\|_2 \\
=\ & (1 - 2/d_{max}) \left\| (\mathbf{H}'[j^*, :] - \mathbf{H}'[i^*, :]) \right\|_2 \|\mathbf{W}_L\|_2
\end{aligned}
\tag{43}
$$

By substituting Eq. 42 and 43 into Eq. 41 based on the connectivity of nodes $i$ and $j$, we can obtain,

$$\frac{2K}{\gamma \cdot m \cdot (N-m)} \sum_{j \in U} \sum_{i \in S} \|\mathbf{H}[j,:] - \mathbf{H}[i,:]\|_2$$

$$\leq \frac{2K}{\gamma \cdot m \cdot (N-m)} \sum_{j \in U} \sum_{i \in S} \begin{cases} \|\mathbf{H}'[j^*,:] - \mathbf{H}'[i^*,:]\|_2 \|\mathbf{W}_L\|_2 & s(i,j) > 1 \\ (1 - 2/d_{max}) \|(\mathbf{H}'[j^*,:] - \mathbf{H}'[i^*,:])\|_2 \|\mathbf{W}_L\|_2 & s(i,j) = 1 \end{cases}$$

$$= \frac{2K}{\gamma \cdot m \cdot (N-m)} \left( m \cdot (N-m) - \frac{2P_1}{d_{max}} \right) \|(\mathbf{H}'[j^*,:] - \mathbf{H}'[i^*,:])\|_2 \|\mathbf{W}_L\|_2$$

$$= \frac{2K}{\gamma} \left( 1 - \frac{2\mathcal{I}_1}{d_{max}} \right) \|(\mathbf{H}'[j^*,:] - \mathbf{H}'[i^*,:])\|_2 \|\mathbf{W}_L\|_2 \quad \text{(Unroll recursion)}$$

$$\leq \frac{2K}{\gamma} \left( 1 - \frac{2\mathcal{I}_L}{d_{max}} \right) \|(\mathbf{X}[j^*,:] - \mathbf{X}[i^*,:])\|_2 \prod_{l=1}^{L} \|\mathbf{W}_l\|_2$$

$$\leq \frac{2K}{\gamma} \left( 1 - \frac{2\mathcal{I}_L}{d_{max}} \right) B_x \prod_{l=1}^{L} \|\mathbf{W}_l\|_2 \tag{44}$$

Thus we have,

$$\ln \mathop{\mathbb{E}}_{f \sim P} \left[ e^{2(m-1)(r_{U,\gamma}(f) - r_{S,\gamma}(f))^2} \right]$$

$$\leq \ln \sup_{f \sim P} \left[ e^{2(m-1)(r_{U,\gamma}(f) - r_{S,\gamma}(f))^2} \right]$$

$$= \ln \sup_{f \sim P} \left[ e^{2(m-1)\left( \frac{2K}{\gamma} \left( 1 - \frac{2\mathcal{I}_L}{d_{max}} \right) B_x \prod_{l=1}^{L} \|\mathbf{W}_l\|_2 \right)^2} \right]$$

$$\leq \sup_{f \sim P} 2(m-1) \left( \frac{2K}{\gamma} \left( 1 - \frac{2\mathcal{I}_L}{d_{max}} \right) B_x \prod_{l=1}^{L} \|\mathbf{W}_l\|_2 \right)^2$$

$$\leq 2(m-1) \left( \frac{2K}{\gamma} \left( 1 - \frac{2\mathcal{I}_L}{d_{max}} \right) B_x B_w^L \right)^2$$

$$\leq O \left( \frac{m K^2 B_x^2 (1 - 2\mathcal{I}_L/d_{max})^2 B_w^{L/2}}{\gamma^2} \right) \tag{45}$$

Now we remains to bound the term $\mathcal{D}_{KL}(Q||P)$ following the procedure in Neyshabur et al. (2017) which derives the generalization bound for ReLU neural networks based on the perturbation bound. Owing to the Lipschitz property of the ReLU function under L2 norm, we can deduce that the gap between the outputs of two nodes under L2 norm is given by:

$$\begin{aligned} \|\mathbf{H}_l[i,:] - \mathbf{H}_l[j,:]\|_2 &= |\sigma(\mathbf{A}\mathbf{H}_{l-1}\mathbf{W}_l)[i,:] - \sigma(\mathbf{A}\mathbf{H}_{l-1}\mathbf{W}_l)[j,:]\|_2 \\ &\leq c\|\tilde{\mathbf{A}}\mathbf{H}_{l-1}\mathbf{W}_l[i,:] - \tilde{\mathbf{A}}\mathbf{H}_{l-1}\mathbf{W}_l[j,:]\|_2 \\ &\leq c\|\tilde{\mathbf{A}}^L \mathbf{X} \prod_{l=1}^{L} \mathbf{W}_l[i,:] - \tilde{\mathbf{A}}^L \mathbf{X} \prod_{l=1}^{L} \mathbf{W}_l[j,:]\|_2 \end{aligned}$$

Thus, when analyzing the generalization of a GCN, if one employs the Lipschitz activation and considers the L2 norm of the gap between the outputs of two nodes, the GCN can be equated to an MLP where the input $\mathbf{X}$ is transformed to $\mathbf{A}^L \mathbf{X}$. This explains why the techniques in Neyshabur et al. (2017) can be adapted to GCN with appropriate modifications.

Let $\beta = \left( \prod_{l=1}^{L} \|\mathbf{W}_l\|_2 \right)^{1/L}$ and normalize the weight of the network as $\tilde{\mathbf{W}}_i = \frac{\beta}{\|\mathbf{W}_i\|_2} \mathbf{W}_i$. Due to the homogeneity of the ReLU, we have $f_{\tilde{\mathbf{w}}} = f_{\mathbf{w}}$. We select the prior $P = \mathcal{N}(0, \sigma^2 I)$ for the vectorized parameters of GNN. The posterior is defined by adding the random perturbation $\mathbf{u} \sim \mathcal{N}(0, \sigma^2 I)$ to the prior. $\sigma$ is set based on an approximation $\tilde{\beta}$ of $\beta$ since the prior cannot depend on the learned network. We will calculate the PAC-Bayes bound for each $\tilde{\beta}$, which holds for all

networks $\mathbf{W}$ with the range $|\beta - \tilde{\beta}| \leq \frac{1}{L}\beta$. In the end, we will then utilize the union bound over all $\tilde{\beta}$ to obtain the cover all possible networks. First we consider a fixed $\tilde{\beta}$ and all networks satisfying $|\beta - \tilde{\beta}| \leq \frac{\beta}{L}$. From Tropp (2012), we get the bound for the spectral norm of $\mathbf{U} \sim \mathcal{N}(0, \sigma^2 I)$,

$$\Pr(\|\mathbf{U}_l\|_2 \geq t) \leq 2he^{-t^2/2h\sigma^2} \tag{46}$$

Thus, for perturbations on all layers, we have,

$$\Pr\left(\bigcup_{l=1}^{L} \|\mathbf{U}_l\|_2 < t\right) \geq 1 - 2Lhe^{-t^2/2h\sigma^2} \tag{47}$$

Setting $1 - 2Lhe^{-t^2/2h\sigma^2} = \frac{1}{2}$, we have at least probability $\frac{1}{2}$, the spectral norm of the perturbation in each layer $\|\mathbf{U}_i\|_2 \leq \sigma\sqrt{2h\ln(4Lh)}$. Plugging this to perturbation bound in Lemma 6, we have with probability a least $\frac{1}{2}$,

$$
\begin{aligned}
\max_{i \in S \cup U} &|f_{\mathbf{w}+\mathbf{u}}(\mathbf{A}, \mathbf{X})[i, :] - f_{\mathbf{w}}(\mathbf{A}, \mathbf{X})[i, :]|_2 \\
\leq\ & eB_x(\prod_{i=1}^{L} \|\mathbf{W}_i\|_2) \sum_{i=1}^{L} \frac{\|\mathbf{U}_i\|_2}{\|\mathbf{W}_i\|_2} \\
=\ & eB_x\beta^L \sum_{i=1}^{L} \frac{\|\mathbf{U}_i\|_2}{\beta} \\
\leq\ & eB_x\beta^{L-1}L\sigma\sqrt{2h\ln(4Lh)} \\
\leq\ & e^2 B_x\tilde{\beta}^{L-1}L\sigma\sqrt{2h\ln(4Lh)} \\
\leq\ & \frac{\gamma}{4}
\end{aligned}
\tag{48}
$$

where in the last inequality we set $\sigma = \frac{\gamma}{42B\tilde{\beta}^{L-1}\sqrt{d\ln(4Lh)}}$. The perturbation $\mathbf{u}$ with the above chosen $\sigma$ satisfy the assumptions of the Lemma 7.

Then we calculate the KL divergence term in Lemma 7.

$$
\begin{aligned}
\mathcal{D}_{KL}(Q\|P) &= \frac{|w|^2}{2\sigma^2} = \frac{42^2 B_x^2 \tilde{\beta}^{2L-2} L^2 h \ln(4Lh)}{2\gamma^2} \sum_{l=1}^{L} \|\mathbf{W}\|_F^2 \\
&\leq O\left(\frac{B_x^2 \tilde{\beta}^{2L} L^2 h \ln(4Lh)}{\gamma^2} \sum_{l=1}^{L} \frac{\|\mathbf{W}\|_F^2}{\beta^2}\right) \\
&\leq O\left(B_x^2 L^2 h \ln(4Lh) \frac{\prod_{l=1}^{L} \|\mathbf{W}_i\|_F^2}{\gamma^2} \sum_{l=1}^{L} \frac{\|\mathbf{W}_l\|_F^2}{\|\mathbf{W}_l\|_2^2}\right)
\end{aligned}
$$

Thus, we have for any $\tilde{\beta}$, we probability at least $1 - \delta$ for all $\mathbf{W}$ with $\beta - \tilde{\beta} \leq \frac{\beta}{L}$,

$$r_{U,0}(f) \leq R_{S,\gamma}(f) + O\left(\sqrt{\frac{B_x^2 L^2 h \ln(4Lh)\Pi\Sigma + \ln\frac{m}{\delta}}{\gamma^2 m}} + K^2 B_x^2 (1 - \frac{2\mathcal{I}_L}{d_{max}})^2 B_w^L\right) \tag{49}$$

where $\Pi = \prod_{l=1}^{L} \|\mathbf{W}_i\|_2^2$ and $\Sigma = \sum_{l=1}^{L} \frac{\|\mathbf{W}_l\|_F^2}{\|\mathbf{W}_l\|_2^2}$ are product and sum of spectral norm of weights. The last step is to take a union bound over all choices of $\tilde{\beta}$ s.t. $|\tilde{\beta} - \beta| \leq \frac{\beta}{L}$. We only need to consider $\beta$ in the range of $\left(0, \left(\frac{\gamma\sqrt{m}}{2B_x}\right)^{\frac{1}{L}}\right]$, since for the $\beta$ our of this range, the bound trivially

holds. If $\beta > \frac{\gamma\sqrt{m}}{2B_x}$, then we have,

$$\frac{B_x^2 L^2 d \ln(4Ld) \prod_{l=1}^{L} \|\mathbf{W}_i\|_F^2 \sum_{l=1}^{L} \frac{\|\mathbf{W}_l\|_F^2}{\|\mathbf{W}_l\|_2^2} + \ln \frac{m}{\delta}}{\gamma^2 m}$$

$$\geq \quad \frac{L^2 h \ln(Lh)}{4} \sum_{l=1}^{L} \frac{\|\mathbf{W}_l\|_F^2}{\|\mathbf{W}_l\|_2^2} \geq 1$$

(50)

which indicates that the bounds trivially holds since $r_{U,0}$ is bounded by 1. A sufficient condition to satisfy $|\tilde{\beta} - \beta| \leq \frac{\beta}{L}$ is $|\tilde{\beta} - \beta| \leq \frac{1}{L}\left(\frac{\gamma}{2B_x}\right)^{\frac{1}{L}}$. And we need at most $Lm^{\frac{1}{2L}}$ to cover all possible $\beta$. Taking a union bound on all such $\beta$, gives the final bound,

$$r_{U,0}(f) \leq R_{S,\gamma}(f) + O\left(\sqrt{\frac{B_x^2 L^2 h \ln(4Lh)\Pi\Sigma + \ln \frac{Lm}{\delta}}{\gamma^2 m}} + K^2 B_x^2 (1 - \frac{2\mathcal{I}_L}{d_{max}})^2 B_w^L\right) \quad (51)$$

$\square$

## C  THE STRUCTURE IMBALANCE PHENOMENON IN GRAPH LEARNING.

After grouping the test nodes into five groups based on their distance from the training nodes, it is observed that the generalization error increases as the distance decreases. Theorem 1 can be used to theoretically explain this phenomenon.

**Test subset construction.** We first select 20 nodes from each class for the training set, 500 nodes for the validation set, and 1000 nodes for the test set. The test nodes are then divide into five groups according to their minimum distance to any nodes in the training set, i.e., $\forall i \in U$, $d(i) = \min_{j \in T}(s(i,j))$. The minimum distances from the nodes in the five test subsets to the training set are $\{1, 2, [3,4], [5,\infty), \infty\}$. interplay with training nodes decreases sequentially within different subsets. Note that the nodes in the first test subset are directly connected to some training nodes, and the nodes in the last test subset are isolated to all training nodes. We train a *two-layer* Vanilla GCN (Kipf & Welling, 2017) in a transductive setting. We repeat the experiment with different node splits 100 times and report the average accuracy on each test subset.

**Experimental results.** Fig. 1 shows that the accuracy on different test subsets decreases as the interplay diminishes. We use a three-layer GNN, allowing any node can directly obtain information from nodes within three-hop neighbors via three times message-passing. Thus the nodes in the first test subset can interact with the training nodes three times, while the second group can interact twice. In the training phase, the features of these test nodes can be indirectly passed into the training objective to affect the model. In the inference phase, the features of training nodes can be utilized when predicting labels for the first two subsets, resulting in better generalization ability. On the last test subset, a significant drop in accuracy is observed, which we attribute to two aspects: 1) No path exists between the test nodes and the training nodes, so no information interaction occurs in both training and inference phases. 2) The difference in features between training nodes and isolated test nodes is high.

**Insights.** The theorem and experimental results provide us the following insights for graph learning:

- **Enhance the interplay between the training and test sets to improve generalization ability..** Transductive learning additionally uses the features of the test set, but the optimization objective of the model is limited to the prediction error on the training nodes. Hence we need to pass test node information to training nodes, enabling the optimization objective to perceive test nodes' patterns. Long-distance message passing in GNNs with multilayer stacking is challenging. Techniques such as residual connections (Li et al., 2019; Chen et al., 2020b), random edge drop (Rong et al., 2020) alleviate concerns, but connecting distant/isolated test nodes remains difficult.

- **Structurally even selection of training data.** The generalization performance significantly declines when test nodes exceeds the training nodes' receptive field in graph convolution. We need select training nodes with a structurally even distribution across the graph to radiate as many nodes

as possible. Considering the link between generalization performance and node degree Tang et al. (2020); Liu et al. (2023), we modify the grouping criterion to node degree, revealing a similar trend (Fig. 2). We attribute this to nodes with smaller degrees exhibit fewer interactions with training nodes.

## D  GATED MECHANISMS IN GLOBAL WORKSPACE

The content of global workspace is updated using a gated approach as in Santoro et al. (2018). $\mathbf{M}^t$ and $\mathbf{M}^{t+1}$ are memory matrix in current and next time step. $\mathbf{I}$ and $\mathbf{F}$ are input and forget gates. The gated mechanism is formulated as follows:

$$
\begin{aligned}
\overline{\mathbf{X}} &= \text{relu}\left(\mathbf{X} \times \mathbf{W}^1\right) \\
\mathbf{K} &= \overline{\mathbf{X}} + \tanh\left(\mathbf{M}^t\right) \\
\mathbf{I} &= \text{sigmoid}\left(\mathbf{KW}^I\right) \\
\mathbf{F} &= \text{sigmoid}\left(\mathbf{KW}^F\right) \\
\mathbf{M}^{t+1} &= \mathbf{I} \times \tanh(\mathbf{M}^t) + \mathbf{F} \times \mathbf{M}^t
\end{aligned}
$$

## E  ALGORITHM OF INTERPGNN

---

**Algorithm 1** Training procedure of InterpGCN-GW in mini-batch manner

---

**Input**: A graph $G = (V, E)$, node set $V = \{1, \cdots, N\}$, edge set $E$, feature matrix $\mathbf{X} = [\mathbf{X_S}, \mathbf{X_U}]$, training labels $Y_S$, GNN encoder $g1, g2$, $\mathbf{W}$ feature embedding dimension $n_d$, transformer layer, node2vec embedding dimension $n_p$, memory matrix $\mathbf{M} = [m_1, \cdots, m_{n_m}]$, batch size $B$ and other hyper-parameters in Appendix F.2.

**Output**: Predicted labels for all nodes $\hat{y}_i$

1: **Preprocess and initialization**:
2: Generate node2vec embedding $X_{pe}^{N \times n_p}$.
3: Partition graphs into $c$ subgraphs with METIS.
4: Initialize the memory in the global workspace.
5: Initialize the parameter of GNN encoder and transformer layer.
6: **for** sample a subgraph $G_c = \{V_c, E_c\}$ from $C$ clusters without replacement. **do**
7:     **Step 1: Extract initial feature embedding and positional encoding**
8:     residual $= \mathbf{H}_c$
9:     $\mathbf{H}_c = g(\mathbf{X}, G)$
10:     $\mathbf{H}_c = [\mathbf{H}_c, \mathbf{X}_{pe}]$ (concatenate positional encoding and feature embedding.)
11:     **Step 2: Nodes compete to write in the global workspace.**
12:     $\mathbf{M}^t = [\mathbf{M}^t, \mathbf{H}]$
13:     $S = \text{softmax}\left(\frac{\mathbf{Q}_w \mathbf{K}_w^\top}{\sqrt{d_k}}\right) = \text{softmax}\left(\frac{\mathbf{M}^t \mathbf{W}_{w,q} (\mathbf{H}_c \mathbf{W}_{w,k})^\top}{\sqrt{d_k}}\right)$
14:     Set $\mathbf{S}$ as its top-$k$ columns (optional)
15:     $\mathbf{M}^{t+1} \leftarrow \mathbf{SV}_w = \mathbf{SH}_c \mathbf{W}_{w,v}$
16:     **Step 3: Global workspace broadcast to all nodes.**
17:     $\widehat{\mathbf{H}_c} = \text{softmax}\left(\frac{\mathbf{H}_c \mathbf{W}_{r,k} (\mathbf{M}^{t+1} \mathbf{W}_{r,k})}{\sqrt{d_k}}\right) \mathbf{M}^{t+1} \mathbf{W}_{r,k}$
18:     $\mathbf{H}_o = \sigma(\mathbf{A_c}([\widehat{\mathbf{H}_c}, \text{residual}] \oplus \sigma \cdot \mathbf{H}_c)\mathbf{W}_o)$
19:     $\hat{y}_i = \text{Softmax}(\mathbf{H}_o)[i]$
20:     $\mathcal{L}_c = \text{NLL\_loss}(\hat{y}_i, y) \forall i \in V_c$
21:     Update the parameters of GNN encoder and all $\mathbf{W}$ in global workspace with AdamW optimizer by minimizing $\mathcal{L}_c$.
22: **end for**
23: **return** The GNN encoder $g_\theta(\cdot)$

---

## F  NODE CLASSIFICATION TASK

### F.1  DATASET DESCRIPTION

Cora, CiteSeer, and Pubmed (Yang et al., 2016), and CoraFull (Bojchevski & Günnemann, 2018) are citation network datasets, where nodes represent documents, edges represent citation links, and

Table 8: Statistics of used datasets.

| Datasets | # Nodes | # Edges | # Features | # Classes |
|---|---|---|---|---|
| structure imbalance experiment | | | | |
| Cora | 2,708 | 10,556 | 1,433 | 7 |
| CiteSeer | 3,327 | 9,104 | 3,703 | 6 |
| Pubmed | 19,717 | 88,648 | 500 | 3 |
| small-scale node classification | | | | |
| CoraFull | 19,793 | 126,842 | 8,710 | 70 |
| Wiki-CS | 11,701 | 216,123 | 300 | 10 |
| Computer | 13,752 | 491,722 | 767 | 10 |
| Photo | 7,650 | 238,162 | 745 | 8 |
| Flickr | 89,250 | 899,756 | 500 | 7 |
| large-scale node classification | | | | |
| Reddit | 232,965 | 114,615,892 | 602 | 41 |
| Aminer-CS | 593,486 | 6,217,004 | 100 | 18 |
| Amazon2M | 2,449,029 | 61,859,140 | 100 | 47 |

features are bat-of-words embeddings. Wiki-CS (Mernyei & Cangea, 2020) is a Wikipedia-based dataset, where nodes represent computer science articles, edges represent hyperlinks, classes represent different branches of the field, and features are bag-of-word embeddings. Computer, Photo from Shchur et al. (2018) and Amazon2M (Chiang et al., 2019) are segments of the Amazon co-purchase graph, where nodes represent goods, edges indicate that two goods are frequently bought together, features are bag-of-words embeddings of product reviews, and class represent product category. Flick from Zeng et al. (2020) is an collection of images, where nodes represent images uploaded to Flickr, edges indicate that two images share some common properties, features are bag-of-words embeddings of the images. Reddit from Hamilton et al. (2017) is a social network dataset, where nodes represent posts on Reddit, edges indicate if the same user comments on both posts, features are bag-of-words embeddings, and labels are communities. Aminer-CS (Feng et al., 2020) is citation network dataset, where nodes represent a paper in computer science, edges represent citation relations among papers, features are bag-of-word vector of paper abstract, and labels are topics of papers.

### F.2 IMPLEMENTATION DETAILS

**Software and Software infrastructures.** Our code is built based on Goyal et al. (2022)[1] and Ott et al. (2019)[2] with:

- Software dependencies: Python 3.7.11, Pytorch 1.9.1, Pytorch-geometric 2.0.1, Numpy 1.20.1, scikit-learn 0.24.2, DGL-cuda10.1 0.7.1, scipy 1.6.2.

- CPU: Intel(R) Xeon(R) Gold 5222 CPU @ 3.80GHz

- GPU: 2 NVIDIA GeForce RTX 3090

- OS: Ubuntu 18.04.6 LTS

**Implementation details** For baseline models, the hyperparameter settings follow their official implementation. We use the node2vec as the positional encoding and GNN for initial feature embedding in InterpGCN-GW. Additionally, the historical embedding in Fey et al. (2021) is added and denoted as InterpGAS-GW. We use the NLL-Loss for multi-class classification. The AdamW (Loshchilov & Hutter, 2017) optimizer is used for gradient decent optimization. We also use the BatchNorm and early stop strategies for all models. For the model configuration of InterpGCN-GW, the selection range of hyperparameters is presented as follows:

- Training details:

    - Learning rate: $\{0.01, 0.001\}$

---

[1]https://github.com/anirudh9119/shared_workspace
[2]https://github.com/facebookresearch/fairseq, MIT license

Table 9: Efficiency comparision on large-scale graphs.

| | Aminer-CS | | | | Reddit | | | |
|---|---|---|---|---|---|---|---|---|
| | batch size = 126600 | | batch size = 63300 | | batch size = 40000 | | batch size = 20000 | |
| | Memory (MB) | Time (s) | Memory (MB) | Time (s) | Memory (MB) | Time (s) | Memory (MB) | Time (s) |
| NAGPhomer | 10310 | 5.1 | 8316 | 4.6 | 9960 | 7.2 | 6920 | 4.1 |
| NodeFormer | 10011 | 1.8 | 5471 | 1.5 | 9043 | 5.2 | 4617 | 4.7 |
| ClusterGCN | 4835 | 2.2 | 4105 | 1.4 | 7793 | 7.5 | 4107 | 5.4 |
| InterpGCN-GW | 8503 | 3.4 | 5391 | 2.8 | 7959 | 9.7 | 4925 | 6.4 |

- – Weight decay: {0.001, 0.0001}
- – Number of partitions for Amazon2M: {5000, 15000, 30000}
- – Number of partitions for Aminer-CS: {3000, 6000, 12000}
- – Number of partitions for Reddit: {600, 1500, 3000}
- – Batch size: {64, 128, 256, 512}
- Node2vec position encoding:
  - – Node2vec embedding dimension: {32, 64, 128, 256}
  - – Walk length: {32, 64, 128, 256}
  - – Context size: {32, 64, 128, 256}
  - – Walks per node: {16, 32, 64}
  - – Batch size: {128, 256, 512}
  - – Walk length: {32, 64, 128, 256}
- GNN feature embedding extractor:
  - – Hidden unit: {32, 64, 128, 256}
  - – Number of layers: {2, 3, 4}
  - – Dropout: {0.0, 0.1, 0.2, 0.5}
  - – Activation: LeakyReLU
- Global workspace:
  - – Memory length : {32, 64, 128, 256, 512}
  - – Number of memory slots: {8, 16, 32, 64}
  - – GW attention embedding dimension: {32, 64, 128, 256, 512}
  - – GW forward fully connected layer dimension: 256, 512
  - – GW number of heads: {2, 4, 8}
  - – GW dropout: {0.1, 0.2, 0.4}
  - – Use LayerNorm: True

# G  ADDITIONAL EXPERIMENTS

## G.1  GPU MEMORY COMPARISON ON LARGE-SCALE GRAPHS

In this section, we validate the efficiency of InterpGCN-GW on large graphs. We draw comparisons with ClusterGCN, NAGPhormer, and NodeFormer in terms of GPU memory usage and training time. The three baselines all use their official implementations. For a fair comparison, we set the number of layers for all models to 2, the number of hidden units to 128, and utilize the same batch size. We report the time required to train for one epoch in Table 9. ClusterGCN can be seen as our method with the Global Workspace component removed. Compared to ClusterGCN, our additional memory requirement is relatively modest, demonstrating the efficiency of use of our

method. The results show that, compared to Graph Transformer, our model consumes less memory. This is because our method reduces the complexity from $O(N^2)$ to $O(Nn_m)$, where $n_m$ is much smaller than $N$. Despite NodeFormer reducing the complexity to linear, at the same batch size, we still use less memory, offering better scalability.

## G.2 FURTHER EXPLANATION ON INTERPLAY WITH INTERPGCN

**Interplay enhancement with global workspace for GCN.** In this section, we first explain why global workspace can enhance the interplay for GCN with following corollary. Note that the GCN with global workspace is an simplification our InterpGCN by removing attention.

**Corollary 1 (PAC-Bayes bounds for GCN with global workspace)** *Let $f \in \mathcal{H}$ be an $L$-layer GCN with parameters $\{\mathbf{W}_i\}_{i=1}^L$, for any $B_x, B_w > 0$, $L, h, K, n_m \geq 1$, and any $\delta, \gamma > 0$, with probability at least $1 - \delta$ over a training set $S$ of size $m$ we have,*

$$r_{U,0}(f) \leq R_{S,\gamma}(f) + O\left(\sqrt{\frac{B_x^2 L^2 h \ln(4Lh)\Pi\Sigma + \ln\frac{Lm}{\delta}}{\gamma^2 m}} + K^2 B_x^2 (1 - \frac{2\mathcal{I}_L + \sum_{r=1}^L r \cdot n_m/m}{d_{max}})^2 B_w^L\right) \quad (52)$$

*where $\Pi = \prod_{l=1}^L \|\mathbf{W}_i\|_F^2$ and $\Sigma = \sum_{l=1}^L \frac{\|\mathbf{W}_l\|_F^2}{\|\mathbf{W}_l\|_2^2}$ are product and sum of spectral norm of weights, $d_{max}$ is the maximum node degree, $n_m$ is the number of memory slots each of which stores feature of an arbitrary node.*

We illustrate above corollary using a specific example with a 3-layer GCN. We select a training set of $N_s$ nodes and $N_u$ test nodes isolated from the training set, where the initial $L$-hop interplay is 0. Then the global workspace can store full features of $n_m$ training nodes, which can be broadcast to all test nodes. This effectively transforms all test nodes into 1-hop neighbors of stored training nodes, connected to at least $n_m$ training nodes. Consequently, the interplay of GCN increases to at least $\frac{6}{N_m}$, leading in a smaller generalization error bound. This finding can be supported with Liu et al. (2022), which use dummy nodes connected to all nodes to boost graph structure learning, since memory slots without attention can act as virtual nodes. Building on this corollary, we use attention to selectively write important node information to memory. Extending Theorem 1 to fully encompass InterpGNN is challenging, as the selection of nodes for memory is real-data-dependent. Furthermore, Theorem 1 factors in model complexity. The introduction of an attention mechanism in InterpGNN could potentially lead to a higher PAC-Bayesian generalization error compared to GCN. Thus, we experimentally validate that InterpGNN enhances interplay

**Visualization of attention scores.** We visualize the attention scores to elucidate the function of InterpGNN in enhancing interplay. We select 40 nodes within the Cora dataset for a clear comparison, using 32 memory slots in our model. Figure 8 illustrates the second step's attention over nodes, indicating preferences for nodes to be stored in memory. Figure 9 visualizes the third step's attention over memory, revealing nodes' preferences in reloading information from memory slots. Figure 10 presents the cosine similarity of node features with heat map and annotates shortest path lengths between node pairs. Our key observations include: 1) Node 34, isolated from other nodes (as seen in Figure 9), cannot engage in message passing in GCN. In our approach, Node 34 accesses information from other nodes via memory slots. 2) Nodes 27 and 16 share high feature similarity and the same label, but are separated by a shortest path length of 9, which impedes direct message passing in a Vanilla GCN. In our approach, Node 27's features are written into the memory, and Node 16 can access this information by reading from the memory. This allows for effective communication between these two nodes despite their distance in the graph structure. 3) Memory tends to store information from Node 26 and 27, maybe because they share relatively high feature similarity with other nodes, marking them as important nodes.

## G.3 FURTHER DISCUSSION ON HETEROPHILIC GRAPHS

Homophily and heterophily are pivotal properties in graph structure, with extensive research dedicated to exploring these concepts Luan et al. (2022); Mao et al. (2023); Luan et al. (2023); Ma et al. (2022). In this section, we delve into the applicability of our derived generalization error and the proposed method to heterophilic graphs.

**Theory perspective.** Theorem 1 can be extended to include heterophilic graphs with an additional assumption. The primary distinction between homophily and heterophilic graphs lies in whether connected nodes belong to the same category. In our theoretical derivation, we assume that the true

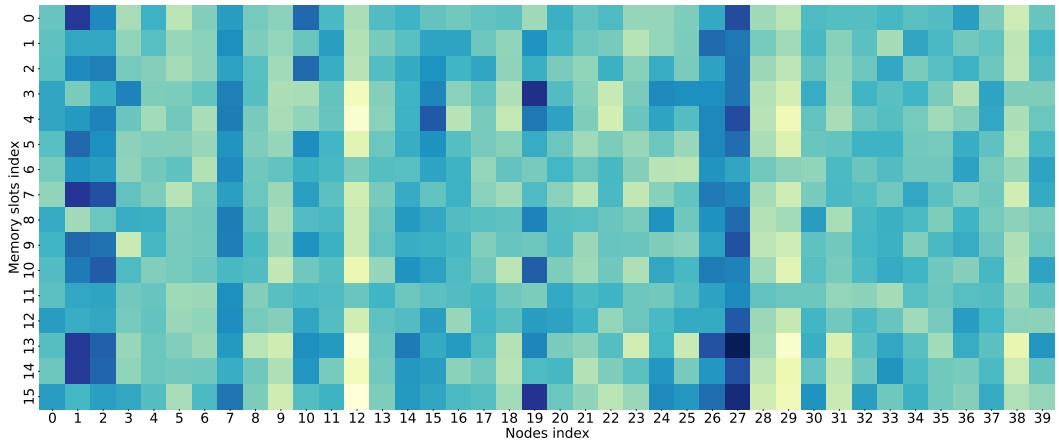

Figure 8: Attention over nodes.

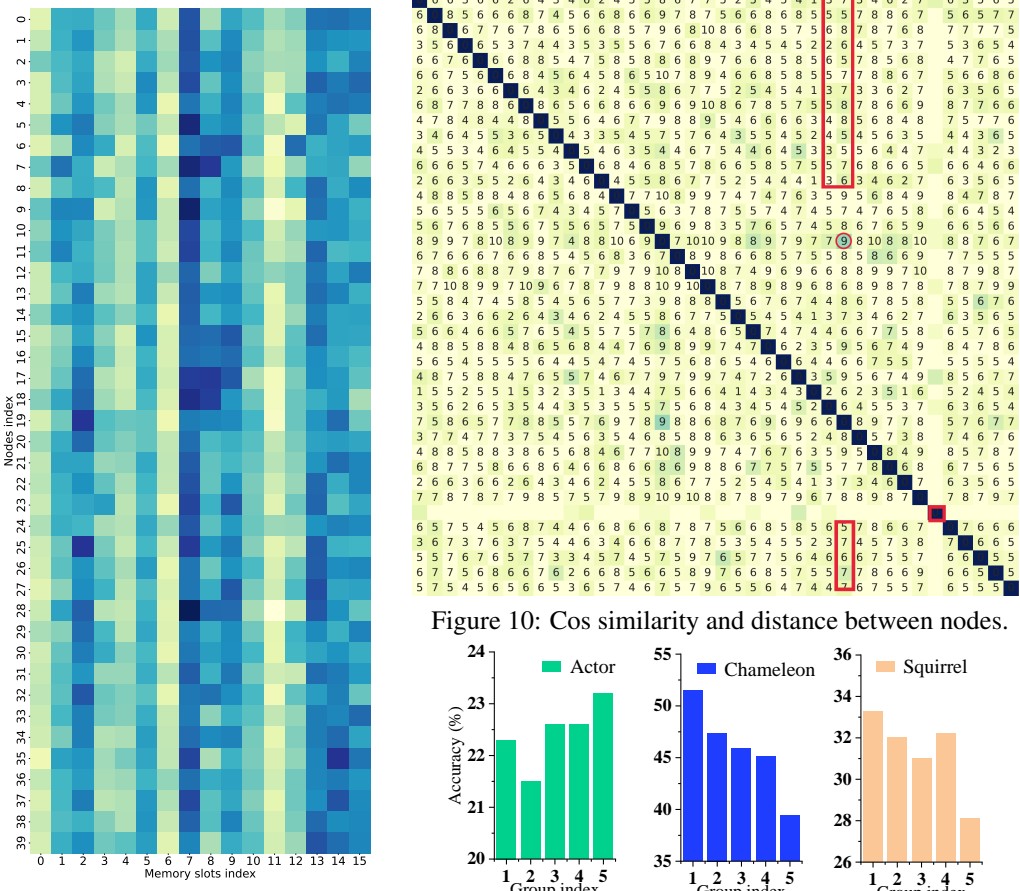

Figure 9: Attention over memory.

Figure 10: Cos similarity and distance between nodes.

Figure 11: Grouping effect on heterophilic graphs.

labels are solely dependent on features, meaning nodes with similar features are likely to be in the same class. Formally, for two nodes $i$ and $j$ with labels $k$, the difference in their label probabilities is related to their feature difference:

$$|P(y_i = k|x_i) - P(y_j = k|x_j)| \leq C|x_i - x_j|,$$

Table 10: Ablation study on memory and the gate control of memory.

| small graphs | CoraFull | Wiki-CS |
|---|---|---|
| InterpGCN | $71.72 \pm 0.37$ | $82.92 \pm 0.18$ |
| w/o memory | $71.05 \pm 0.24$ | $81.01 \pm 0.21$ |
| w/o gate | $71.31 \pm 0.22$ | $82.23 \pm 0.17$ |
| Reddit | Aminer-CS | Amazon2M |
| $93.35 \pm 0.20$ | $68.52 \pm 0.31$ | $89.11 \pm 0.14$ |
| $90.47 \pm 0.41$ | $66.29 \pm 0.28$ | $87.81 \pm 0.24$ |
| $89.49 \pm 0.35$ | $67.21 \pm 0.34$ | $88.42 \pm 0.18$ |

Table 11: Node classification on heterophily graphs (% + std).

| | Actor | Squirrel | Chameleon |
|---|---|---|---|
| GCN | $28.92 \pm 1.67$ | $57.21 \pm 1.66$ | $62.12 \pm 1.74$ |
| NodeFormer | $29.93 \pm 1.78$ | $49.12 \pm 1.22$ | $58.73 \pm 1.84$ |
| InterpGCN-GW | $31.57 \pm 1.29$ | $60.12 \pm 1.31$ | $64.27 \pm 1.96$ |
| H2GCN (Zhu et al., 2020) | $35.21 \pm 2.45$ | $40.79 \pm 1.42$ | $60.13 \pm 1.78$ |
| GGCN (Yan et al., 2022) | $36.79 \pm 1.97$ | $52.19 \pm 1.41$ | $66.19 \pm 2.73$ |
| Diag-NSD(Bodnar et al., 2022) | $36.11 \pm 1.51$ | $52.29 \pm 1.58$ | $62.13 \pm 2.01$ |
| ACM-GCN(Luan et al., 2022) | $33.89 \pm 1.37$ | $56.79 \pm 1.67$ | $67.73 \pm 2.15$ |
| InterpGGCN-GW | $37.81 \pm 2.17$ | $55.49 \pm 1.92$ | $67.17 \pm 1.54$ |
| InterpH2GCN-GW | $35.42 \pm 1.79$ | $41.46 \pm 1.83$ | $59.79 \pm 1.97$ |
| InterpDigNSD-GW | $34.81 \pm 1.43$ | $53.90 \pm 2.17$ | $64.38 \pm 1.43$ |
| InterpACM-GCN-GW | $35.79 \pm 0.84$ | $55.43 \pm 1.60$ | $65.89 \pm 1.67$ |

where $C$ is a constant determined by the true distribution. In heterophilic graphs, we can add a heterophilic coefficient related to node distance to this assumption:

$$|P(y_i = k|x_i) - P(y_j = k|x_j)| \leq C \cdot h_r^{(ij)} \cdot |x_i - x_j|,$$

where $r$ is the distance between nodes $i$ and $j$. In homophily graphs, $h_1$ is small, indicating a tendency for neighboring nodes to be in the same class. Conversely, in heterophilic graphs, $h_1$ is larger, implying neighboring nodes are likely to belong to different classes. Similar to our defined L-hop interplay between training set $S$ and test set $U$, we can define L-hop heterophily as:

$$\mathcal{H}_L = \frac{\sum_{r=1}^{L}(L - r + 1) \cdot |h_r^{i,j}|}{|S| \cdot |U|},$$

where $i \in S$, $j \in U$. In the final generalization error, an additional term on $\mathcal{H}_L$ is included. Taking the dating network dataset from (Zhu et al., 2021) as an example, adjacent nodes are more likely to belong to different classes. Here, due to typically small interplay values, L-hop heterophily $\mathcal{H}_L$ has a more significant impact on generalization error than L-hop interplay $(1 - I_L)$. For non-adjacent nodes, as heterophily is usually not strong (will be discussed later in method perspective), L-hop interplay remains the dominant factor. Regarding the PAC-Bayesian generalization bound in (Mao et al., 2023), a similar approach to heterophily is introduced. However, (Mao et al., 2023) makes strong assumptions about graph datasets, limiting the extendibility of the derived generalization error bounds. Their generalization error is based on that graph is generated with a contextual stochastic block model (CBSM), assuming features of different classes originate from distinct normal distributions, with different connection probabilities $p$ and $q$ for inter-class and intra-class nodes, respectively. The final generalization error bound based on (Ma et al., 2021) includes the homophily ratio difference between training and test sets.

**Method perspective.** The statement regarding the impact of information interaction on generalization error can be amended for heterophilic graphs as follows, more information interaction between non-neighboring test and training nodes can lead to a smaller generalization error for heterophily graphs. This concept aligns with findings from a survey on heterophily in Graph Neural Networks (Zheng et al., 2022). The survey indicates that in heterophilic graphs, nodes sharing high-level structural and semantic similarities may be distant. It also suggests that the representation ability of heterophilic GNNs can be greatly enhanced by focusing on informative features from distant nodes. These elucidate why our method by writing important nodes into the memory and broadcasting them to all nodes, exhibits superior performance on heterophilic graphs. Similarly, many GNNs aggregate information from non-neighboring neighbors. For instance, MixHop (Abu-El-Haija et al., 2019), H2GCN Zhu et al. (2020) propose aggregating information from higher-order neighbors at each message passing step. Similarly, UGCN (Jin et al., 2021) utilizes two-hop networks for message passing, while GGCN (Yan et al., 2022) employs cosine similarity to send signed neighbor features under constraints of relative node degrees. Luan et al. (2022) propose the adaptive channel mixing by adaptively combining different channels of information processing for each node to effectively handle diverse graph structures. The discussion in Luan et al. (2023) does not conflict with these statements. Luan et al. (2023) focuses on the influence of neighboring nodes' label distribution, which necessitates considering the inter-class distinguishability of neighbors. In the example provided in Luan et al. (2023), if two nodes have 2 and 4 neighboring nodes with labels (2, 3) and (2, 2, 3, 3), they would still be classified similarly after aggregation. This supports the necessity of aggregating non-neighboring node features, consistent with our method's approach.