# OpenReview forum: "InterpGNN: Understand and Improve Generalization Ability of Transdutive GNNs through the Lens of Interplay between Train and Test Nodes"
_ICLR.cc/2024/Conference — ICLR 2024 poster_

### Official Review · Reviewer_qP57 · 2023-10-30

**Soundness:** 3 good
**Presentation:** 3 good
**Contribution:** 2 fair
**Rating:** 6
**Confidence:** 3

**Summary:**

This paper proposes a PAC-Bayesian generalization error bound for GNNs and shows the influence of the interplay between training and testing sets on the generalization ability. Based on the theoretical analysis, the authors design a plug-and-play Graph Global Workspace module to enhance the generalization capability.

**Strengths:**

1. The PAC-Bayesian generalization error bound for GNNs is interesting and it shows how the interplay between training and testing sets influences the generalization ability.

2. The proposed method outperforms most baseline methods across multiple graph datasets.

3. The presentation of this paper is good and the paper is easy to follow.

**Weaknesses:**

1. Although the authors derive PAC-Bayesian bound for GNNs in the transductive setting and show how the interplay between training and testing sets influences the generalization ability, I fail to see the strong connection between the theoretical analysis and the proposed method. The proposed method seems to simply adopt the idea of the self-attention mechanism from the transformer and apply it to the graph. It's not clear to me how the proposed method enhances the generalization for the distant nodes.

2. My major concern about the proposed method is the graph partition as partitioning the graph usually leads to information loss. Though node2vec is used for positional encoding purposes, it only encodes the local topological structure, and it cannot compensate for the lost information between different subgraphs. Based on algorithm 1 in Appendix E, there is no information exchange between different subgraphs. The nodes in a subgraph can only receive the information from other nodes within this subgraph and these nodes are isolated from the nodes from other subgraphs. The performance seems to highly depend on the quality of the graph partition algorithms. However, it's unclear whether different graph partitions will influence the performance of the proposed method or not.

3. Some experimental setups are not quite clear. See questions below.

**Questions:**

1. In equation 7, why do you need max operation since $L-r+1>0$ always holds?

2. In Figures 1 and 2, the group 2 usually have the higher accuracy and smaller generalization error. Do you have any explanation for this observation?

3. In the experiment on comparison with GNN variants using Dummy nodes, what is the method (denoted as Dummy nodes in Figure 7) used for comparison? This experiment is confusing.

---

> ### Author Response · Authors · 2023-11-18
> **Response to Reviewer qP57 (Part 1)**
>
> We thank you for recognizing the significance of our derived generalization bound and its insightful implications for GNNs, as well as your positive remarks on our method's performance and the paper's presentation. Below are our detailed response.
>
> **Q1:. The proposed method seems to simply adopt the idea of the self-attention mechanism from the transformer and apply it to the graph. It's not clear to me how the proposed method enhances the generalization for the distant nodes.**
>
> **R1**: We wish to clarify that our method extends beyond merely applying the self-attention mechanism to graphs. Unlike traditional graph transformers that computes node-to-node attention, our approach calculates attention between nodes and a memory component. This memory stores crucial node features and then broadcasts this information to all nodes, acting as a bridge for transferring information across distant nodes. This is particularly significant for large-scale graphs in mini-batch training, where the memory enables efficient information flow between out-of-batch nodes, thus enhancing the overall connectivity and information dissemination in the graph neural network.
>
> **Q2: Based on algorithm 1 in Appendix E, there is no information exchange between different subgraphs. The nodes in a subgraph can only receive the information from other nodes within this subgraph and these nodes are isolated from the nodes from other subgraphs. The performance seems to highly depend on the quality of the graph partition algorithms. However, it's unclear whether different graph partitions will influence the performance of the proposed method or not.**
>
> **A2**: Thank you for your insightful observations regarding the challenges of graph partitioning and the potential for information loss in scalable GNNs and Graph Transformers. Your major concern highlights a key advantage of our method.  It appears  there is a slight misunderstanding about our method. We would like to clarify again that our method does not simply rely on node-to-node attention mechanisms within individual batches. Rather, we employ an attention mechanism to select critical nodes and store their embeddings into a workspace. This workspace is accessible during each batch's training, enabling out-of-batch nodes to indirectly exchange information. This aspect is particularly beneficial for large graphs, as it enhances out-of-batch node information interaction, overcoming limitations of existing methods.
>
> Moreover, as you have  pointed out, the performance of our method does exhibit a certain level of dependency on the graph partitioning strategy employed. To address this, we have conducted a comparison of various partitioning and sampling methods. The results, which we've detailed in the following table, demonstrate that METIS provides the most effective outcomes:
>
> |  | Reddit | Aminer-CS | Amazon2M |
> | --- | --- | --- | --- |
> | random | 91.85 $\pm$ 0.22 | 67.45 $\pm$ 0.41 | 87.19 $\pm$ 0.25 |
> | neighbor sampler | 92.11 $\pm$ 0.29 | 66.89 $\pm$ 0.34 |  88.12 $\pm$ 0.31 |
> | METIS | 93.35 $\pm$ 0.20 | 68.76$\pm$ 0.31  | 89.11$\pm$ 0.14 |
>
>
> **Q3: Why need max operation since $L-r+1 > 0$ always holds?**
> **A3:** The max operation is indeed a typo and will be removed.
>
> **Q4: In figures 1 and 2, the group 2 usually have the higher accuracy and smaller generalization error.  Any explanation for this observation?**
>
> The accuracy observed on real dataset is not entirely equivalent to the generalization error discussed in Theorem 1. A generalization bound is a theoretical limit on a model's expected performance with unseen data, whereas test accuracy is an empirical measure obtained by evaluating the model on a fixed test dataset. While smaller generalization bounds suggest potential for higher accuracy, as indicated by the downward trend in Figure 1, they do not guarantee higher accuracy. The actual accuracy depends on factors like model complexity, training data quality and quantity, and the model's alignment with the data distribution.
>
> Additionally, regarding Figure 1 (test nodes grouped by interplay, as analyzed in our theorem), Group 2 only shows higher accuracy than Group 1 across all models on the CiteSeer dataset. In contrast, on the Cora dataset, all four baseline models perform better in Group 1 than in Group 2. This variation can be attributed to the differing levels of homophily in these datasets. Specifically, CiteSeer exhibits weaker homophily, meaning that adjacent nodes are less likely to share the same label. The homophily of a node $i$ is defined as the ratio of neighbor nodes with the same label:
>
> $$ h_{i}=\frac{|\{j \in \mathcal{N}_ {i}: y_{i}=y_{j} \} |}{d_{i}} $$
>
> Thus, the overall homophily of the dataset is:
>
> $$ h = \frac{\sum_ {i \in V} h_{i}}{|V|} $$
>
> The homophily of Cora dataset is 0.81 and the homophily of CiteSeer dataset is 0.74.

---

> ### Author Response · Authors · 2023-11-18
> **Response to Reviewer qP57 (Part 2)**
>
> **Q5: What is the method denoted as Dummy nodes used?**
>
> **A5:** Dummmy nodes are introduced in [1] to capture global graph properties and boosting graph structure learning. Dummy nodes are initialized with a zero vector and conneted to all other nodes in the graph. Dummy nodes can then aggregate information from all nodes.
>
> The reason we compared our method with dummy nodes is that our memory in InterpGNN-GW acts somewhat like a dummy node. But instead of gathering information from all nodes like a dummy node does, our model uses attention to pick and store features from important nodes into the memory. Also, in mini-batch training, dummy nodes cannot get information from nodes in all batches, which our method can do by using the global workspace.
>
> [1] Liu, Xin, et al. Boosting graph structure learning with dummy nodes. ICML 2022.
>
> ---
> We hope these explanations adequately address your concerns, and we are committed to further refining our paper based on your insightful feedback.

---

> ### Author Response · Authors · 2023-11-22
>
> Thank you once again for your thorough and insightful feedback. We have endeavored to address all your concerns in our responses. We hope that we have clarified any misunderstandings about our method. As we near the end of this discussion phase, we are eager to know if our explanations have satisfactorily addressed your points.
>
> If you have any more comments or questions about our rebuttal, we strongly welcome your feedback. Your guidance is crucial for improving our work, and we look forward to any additional thoughts you might have.

---

> > ### Comment · Reviewer_qP57 · 2023-11-22
> > **Response to Authors**
> >
> > Thank the authors for the further clarifications and the additional experiments! All my concerns are addressed and I would like to increase my score to 6.

---

> > > ### Author Response · Authors · 2023-11-23
> > >
> > > Thank you for acknowledging the additional experiments and for raising your score. We greatly appreciate your constructive feedback, which has been invaluable in strengthening our paper. Your support and recognition are very encouraging.

---

### Official Review · Reviewer_fk5N · 2023-10-31

**Soundness:** 3 good
**Presentation:** 3 good
**Contribution:** 3 good
**Rating:** 6
**Confidence:** 3

**Summary:**

In this paper, the authors derive a PAC-Bayesian bound specifically tailored for message-passing Graph Neural Networks (GNNs) within the context of transductive node classification. They present a novel approach called Graph Global Workspace (InterpGNN-GW), designed to enhance the interaction between nodes both within and across batches. In this method, nodes equipped with positional encoding engage in a competitive process, allowing them to write to and read from a shared global workspace using key-value attention mechanisms. The results of their experiments, conducted on both small- and large-scale graph datasets, reveal the remarkable effectiveness of InterpGNN-GW in node classification tasks when compared to other scalable GNNs and graph transformers. This signifies a noteworthy advancement in the field of graph-based machine learning.

**Strengths:**

1.	They have introduced a PAC-Bayes bound for message-passing Graph Neural Networks (GNNs) in the context of semi-supervised transductive learning. This bound sheds light on the quantifiable relationship between test and training nodes and its impact on the generalization capacity of GNNs in node classification tasks, making it a noteworthy contribution.

2.	Their approach demonstrates superior performance over advanced baseline methods in node classification tasks across a range of dataset scales, thereby highlighting its efficacy and scalability. Moreover, the article exhibits a well-structured and in-depth exploration of the background and related research concerning the generalization bounds of Graph Neural Networks (GNNs). Remarkably, the authors skillfully integrate the utilization of the graph global workspace with the PAC-Bayes bound they have developed in practical applications, offering a significant advantage in terms of global consistency compared to conventional graph transformers.

3.    The comparative experiments are well thought out, taking into account factors such as GNN variants with the inclusion of dummy nodes and historical embeddings.

**Weaknesses:**

1.	The idea of the Global Workspace is intriguing, but it has already been extensively explored in previous research mentioned by authors, diminishing the novelty of this work.
2.	Some concepts lack a comprehensive explanation, such as W_{r,k} in Eq. (10). Further refinement is needed for these elements, including a clearer explanation of the dimensions of linear layers, which would enhance the clarity of the paper.
3. Table 5 does not provide standard deviations.
4. Results provided in table 3 and 4 are limited.

**Questions:**

1.	In Figure 2, the criteria for grouping based on node degree are somewhat confusing. For example, what do the numerical values on the x-axis in the figure represent? I assume these values are simply categories for grouping, rather than actual node degrees. I believe the authors could provide a clearer explanation of the grouping criteria.

---

> ### Author Response · Authors · 2023-11-18
> **Response to Reviewer fk5N (Part 1)**
>
> We thank the reviewer for their thorough review and recognition of the strengths in our work, especially regarding the novelty of the derivied bounds and performance of proposed method. Below are our detailed response.
>
> **Q1: The idea of the Global Workspace is intriguing, but it has already been extensively explored in previous research mentioned by authors, diminishing the novelty of this work.**
>
> **A1**: We are grateful for the reviewer's interest in the Global Workspace concept. Indeed, while the general idea of Global Workspaces has been explored in various fields, our application within the realm of graph analysis introduces novel contributions:
>
> In previous approaches, the Global Workspace primarily served to ensemble different specialized modules. These modules, often pre-trained neural networks, are designed for diverse functions such as sensory perception (including visual or auditory classification, object segmentation), long-term memory storage, reinforcement learning  agents. The interaction of these modules' outputs with the workspace facilitated the extraction of higher-level features.
>
> In contrast, our method utilizes the Global Workspace to store critical node features, addressing specific challenges in graph analysis. Our primary focus is on enhancing the interplay between distant nodes and facilitating inter-batch node information exchange. This approach effectively mitigates the issue of excessive node aggregation often encountered in node-pair Graph transformers. Additionally, considering that GNNs often employ mini-batch training methods on large-scale graphs, leading to a loss of information between different batches, our method facilitates cross-batch information exchange via global workspace, thereby boosting performance on large-scale graphs.
>
> **Q2: Some concepts lack a comprehensive explanation, such as W_{r,k} in Eq. (10). Further refinement is needed for these elements, including a clearer explanation of the dimensions of linear layers, which would enhance the clarity of the paper.**
>
>
> **A2**: We apologize for the confusion caused by a typo in Eq. (10) of our paper. The correct equation should be:
>
> $$
> \widehat{\mathbf{H}_ c}=\operatorname{softmax}\left(\frac{\mathbf{H}_ c \mathbf{W}_ {r, q}\left(\mathbf{M}^ {t+1} \mathbf{W}_ {r, k}\right)}{\sqrt{d_ {k}}}\right) \mathbf{M}^ {t+1} \mathbf{W}_ {r, v}
> $$
>
> We now provide a detailed explanation of the parameters in Eq. (9), Eq. (10), and Eq. (11). The node feature matrix is denoted as $X \in \mathbb{R}^ {N\times d}$, where $N$ is the number of nodes, and $d$ is the dimension of node features. An arbitrary GNN transforms these features into $H \in \mathbb{R}^ {N\times n _f}$, and we obtain the positional embedding $H _{pe} \in \mathbb{R}^{N\times n _p}$. These are concatenated to form the input to the global workspace, which consists of $n _m$ memory slots, each a vector of length $l _m$, represented as $\mathbf{M} = [m _1, \cdots, m _{n _m}]$.
>
> When writing node features to the memory with Eq. (9), keys and values are obtained by multiplying the node embedding with two different linear layers, i.e., $\mathbf{K} _w=\mathbf{H} _c \mathbf{W} _{w,k}$ and $\mathbf{V} _w=\mathbf{H} _c \mathbf{W} _{w,v}$, where $\mathbf{W} _{w,k}$ and $\mathbf{W} _{w,v}$ have input dimensions of $(n _f + n _p)$ and output dimensions of $d _k$ and $d_v$, respectively. The subscript $w$ indicates these values and linear layers are used when writing node features to the Global workspace. The query $\mathbf{Q_w}$ is obtained by multiplying the memory with a linear layer $\mathbf{W} _{w,q}$, whose input dimension is $l_m$ and output dimension matches that of the keys' linear layer. The resulting matrix multiplication of keys and query gives the node-memory attention scores $\mathbf{Q} _w \mathbf{K} _w ^\top \in \mathbb{R}^ {N\times n _m}$. After applying Softmax to these attention scores, they are multiplied with $\mathbf{V} _w$ to select important nodes for updating the memory.
>
> Eq. (10) involves similar linear layers as Eq. (9), with the difference being the swapping of positions between memory and node embedding, to reload contents from the memory back to the nodes. Therefore, the output dimensions of $\mathbf{W} _{r,k}$, $\mathbf{W} _{r,v}$, and $\mathbf{W} _{r,q}$ remain $d_k, d_v, d_k$, respectively, while their input dimensions change to $l_m, l_m, n_f+n_p$.
>
> In Eq. (11), an additional graph convolution operation is performed, where $\mathbf{A} \in \mathbb{R} ^{N\times N}$, $\widehat{\mathbf{H} _c}  \oplus \mathbf{H} _c \in \mathbb{R} ^{N\times (n_f+n_p)}$, and $\mathbf{W} _o \in \mathbb{R} ^{(n_f+n_p)\times K}$, with $K$ being the number of classes.

---

> ### Author Response · Authors · 2023-11-18
> **Response to Reviewer (Part 2)**
>
> **Q3: Table 5 does not provide standard deviations.**
>
> **A3**: The standard deviations derived from the ablation study are similar to those observed in classification tasks using our model. We have included standard deviations in Table 5 in the revised version.
>
> **Q4: Results provided in table 3 and table 4 are limited.**
>
> **A4**: For small graphs, it is indeed true that the improvements by our model are not as prominent. This is primarily due to the limited size of these datasets, which inherently makes it challenging for any model to demonstrate significant improvements. However, it is worth noting that our method still outperforms the baseline models on small datasets. On the other hand, the performance enhancement of our method is  notable on large graphs. The reason for this improvement is due to the out-of-batch node communication via global workspace.
>
>
> Additionally, following Reviewer mAhi's suggestion, we conducted tests in a non-rigorous inductive learning setting. In this scenario, where the model doesn't have access to test node features during training, our method effectively broadcasts information from the training phase to test nodes. Our methods significantly mitigates the performance drop observed in other models when transitioning from a transductive to an inductive learning setting. Our method can achieve close results to those in the transductive setting. For a comprehensive overview of experiments on non-rigorous inductive setting, please refer to the A5 to Reviewer mAhi.
>
>
> **Q5: What do numerical values on the x-axis in the figure represent?**
>
> **A5**: In Figure 2, test nodes are sorted by node degree in descending order and evenly divided into five groups. The x-axis represents these groups, with lower indexes indicating higher node degrees. Specifically, Group 1 (smallest index) contains nodes with the highest degrees, and Group 5 (largest index) comprises those with the lowest.
>
>
>
> ---
> We hope that our responses adequately address the concerns and questions raised. We are grateful for the opportunity to refine our paper based on your insightful feedback.

---

> ### Author Response · Authors · 2023-11-22
>
> Thank you once again for your thorough and insightful feedback. We have endeavored to address all your concerns in our responses. As we near the end of this discussion phase, we are eager to know if our explanations have satisfactorily addressed your points.
>
> If you have any more comments or questions about our rebuttal, we strongly welcome your feedback. Your guidance is crucial for improving our work, and we look forward to any additional thoughts you might have.

---

### Official Review · Reviewer_kZmE · 2023-10-31

**Soundness:** 3 good
**Presentation:** 3 good
**Contribution:** 3 good
**Rating:** 6
**Confidence:** 4

**Summary:**

In this paper, the authors derive transductive PAC-Bayesian bounds for GNNs, by which they reveal that the interplay between training and testing nodes effects the generalization ability of GNNs. With this observation in mind, they design a plug-and-play module to enhance this interplay. This module maintains a memory bank of node embedding and adopts key-value attention mechanism to update the embedding and the model weights. Their framework is validated on both small- and large-scale graph datasets.

**Strengths:**

- The theoretical results are interesting and provide new insights to the generalization of GNNs.
- The proposed module is reasonable, and the experimental results are sufficient to demonstrate its effectiveness.

**Weaknesses:**

- There is a discrepancy between theory and practice. The ''transductive'' or ''semi-supervised'' setting the authors adopt is the same as that in [1], where the randomness come from the labels of nodes. However, in real-world applications, including the experiments of this paper, both nodes features and their labels are fixed, and the randomness come from the random partition of training and test nodes [2,3,4]. Therefore, I think that the authors should use the results in literatures [5,6], e.g., Corollary 7 in [6] to derive similar results as Lemma 1. This can be done by incorporating the techniques in works [7,8]. Otherwise, the authors should conduct some synthetic experiments that treating labels as random variables to verify their theoretical results.
- The authors claim that the proposed module is plug-and-play, which means that it can be applied on any GNNs. However, they only equip GCN with this module. Performing more experiments that applying this module to other type of GNNs is encouraged.

[1] Ma et al, Subgroup generalization and fairness of graph neural networks. NeurIPS 2021.

[2] Oono et al, Optimization and generalization analysis of transduction through gradient boosting and application to multi-scale graph neural networks. NeurIPS 2020.

[3] Esser et al, Learning theory can (sometimes) explain generalisation in graph neural networks. NeurIPS 2021.

[4] Cong et al, On provable benefits of depth in training graph convolutional networks. NeurIPS 2021.

[5] Derbeko et al, Explicit learning curves for transduction and application to clustering and compression algorithms. JAIR 2004.

[6] Begin et al, PAC-Bayesian theory for transductive learning. AISTATS 2014.

[7] Neyshabur et al, A PAC-Bayesian approach to spectrally-normalized margin bounds for neural networks. ICLR 2018.

[8] Liao et al, A PAC-Bayesian approach to generalization bounds for graph neural networks. ICLR 2021.

**Questions:**

- The experiments show that the proposed module could also reduce structural unfairness. Can you provide a further explanation for this phenomenon? If the reason is the interplay between training and testing nodes induced by attention mechanism, I guess that graph transformer models can also reduce structural unfairness.
- Graph Transformer models (e.g., GraphGPS) could also enhance the interplay between training and testing nodes more or less. Why these methods perform inferior to the proposed method?

---

> ### Author Response · Authors · 2023-11-18
> **Response to Reviewer kZmE (Part 1)**
>
> We thank the reviewer for their insightful comments and for acknowledging the theoretical contributions and the effectiveness of our proposed module.
>
> **Q1: The randomness should come from the random partition of training and test nodes instead of labels. Otherwise, the author should conduct some synthetic experiments.**
> **A1**: In response to the reviewer's comments on the use of randomness in our approach, we provide the following clarifications and results:
>
> Firstly, PAC-Bayes, as a method for analyzing generalization error, fundamentally assumes that with probability $1-\delta$, data are sampled from a fixed but unknown distribution, and that model parameters are also drawn from a distribution. Hence, randomness originates from the sampling of data and model parameters. Generalization error is measured through the divergence between the model's performance on prior (before see training data) and posterior probabilities. This assumption is the most commonly used in PAC-Bayes bounds, and these bounds have been experimentally validated on real datasets [9-13]. Without loss of generality, assuming data is sampled from a true distribution $(x,y)\sim D$, we fix $x$ and assume labels are sampled from a real probability distribution $y\sim Pr(x|G)$, following [1]. Works [2,3,4] do not use the PAC-Bayes framework, hence they operate under different assumptions than our theorem.
>
> Secondly, in the context of PAC-Bayes analysis, the randomness coming from the labels of nodes is equivalent to randomness coming from the partition of training and test nodes, since it is assumed that data are sampled from a true distribution. Since the partition follows a uniform distribution, adding a uniform distribution on the data distribution remains invariant. In [6], although the analysis is for transductive learning and assumes randomness from partitioning, the bounds are derived by exhaustively enumerating the number of classification errors $K$ across all $N$ samples (training + test), i.e., $\text{bounds} = \max _{K=0 \ldots N} (\cdot)$, where $N$ is the total number of samples. Then the error of training set follows a hypergeometric distribution.  The maximum generalization error necessarily occurs in the scenario where all classifications are incorrect, which results in trivial bounds.
>
> Thirdly, following your suggestion, we conducted experiments on a synthetic dataset. We used the Stochastic block model implemented with PyTorch Geometric to generate a graph dataset. The node features of each block are sampled from normal distributions, with the centers of clusters being vertices of a hypercube. Due to the memory limitation of 128GB, we could not generate large-scale datasets with the CBM. We created a graph with 6 categories of nodes, 1000 nodes each, where the probability of connections within the same category was set at 0.02 and 0.01, and the probability of connections between different categories was set at 0.001. The other settings remained consistent with those in the paper, and the experimental results are as follows:
>
> | same-class prob | 0.02 | 0.01 |
> | --- | --- | --- |
> | GCN | $29.11 \pm 0.25$ | $26.51 \pm 0.29$ |
> | ClusterGCN | $30.19 \pm 0.21$ | $25.76 \pm 0.33$ |
> | NodeFormer | $30.93 \pm 0.31$ | $27.12 \pm 0.20$ |
> | InterpGCN | $31.89 \pm 0.18$ | $27.87 \pm 0.27$ |
>
> **Q2: Performing more experiments that applying this module to other type of GNNs is encouraged.**
>
> **A2**: In response to your suggestion to demonstrate the plug-and-play nature of our proposed module by applying it to various types of Graph Neural Networks (GNNs), we have expanded our experiments beyond GCN. It is important to note that in our paper, we have already replaced GCN with GnnAutoScale (GAS). Following this, we have integrated our module with other GNN architectures including GAT, GCNII, and SGC. The results of these experiments on various datasets are as follows:
>
> |  | Reddit | Aminer-CS | Amazon2M |
> | --- | --- | --- | --- |
> | InterpGAT-GW | 91.97 $\pm$ 0.29 | 66.87 $\pm$ 0.34 | 85.81 $\pm$ 0.17 |
> | InterpGCNII | 92.43 $\pm$ 0.17 | 67.66 $\pm$ 0.35 | 87.58 $\pm$ 0.28 |
> | InterpSGC | 93.33 $\pm$ 0.24 | 68.41 $\pm$ 0.27 | 88.79 $\pm$ 0.22 |
> | InterpGCN | 93.35 $\pm$ 0.20 | 68.76$\pm$ 0.31 | 89.11 $\pm$ 0.14 |
> | InterpGAS | 92.75 $\pm$ 0.23 | 69.01$\pm$0.22 | 88.59$\pm$ 0.19 |
>
> Additionally, following the suggestion of Reviewer wkbK, we conducted tests on Heterophily graph datasets. We embedded our global workspace module into heterophily graph neural networks, which resulted in performance improvements. Please see details in A8 to reviewer wkbK.

---

> ### Author Response · Authors · 2023-11-18
> **Response to Reviewer kZmE (Part 2)**
>
> **Q3: Explanation for reducing structural unfairness. If the reason is the interpaly between training and testing nodes induced by attention mechanism, graph transformers models can also reduce structural unfairness.**
>
> **A3**: The reduction in structural unfairness in our model is attributed to the enhanced interplay between training and testing nodes facilitated by the workspace. Similarly, graph transformer models are indeed capable of reducing structural unfairness through their attention mechanisms. The generalized error bound we derived is not exclusively limited to our method; rather, we aim to use Theorem 1 to explain the structural unfairness phenomenon in message-aggregating GNN. This also theoretically analyzes why methods that strengthen interplay, such as deep GCN and implicit connections (e.g., graph transformers), can enhance generalization performance.
>
> **Q4: Graph Transformer models (e.g., GraphGPS) could also enhance the interplay between training and testing nodes more or less. Why these methods perform inferior to the proposed method?**
> **A4**: You are indeed correct that graph transformer models can enhance the interplay between training and testing nodes. The proposed method, InterpGCN-GW, surpasses graph transformer models primarily due to two reasons:
>
> **Out-of-Batch Interplay**: InterpGCN-GW can facilitate inter-batch interplay. Graph transformers on large-scale graph datasets typically compute in-batch attention, which limits access to out-of-batch nodes. In contrast, the workspace in our method can be broadcast to nodes across all batches, enhancing interplay among inter-batch nodes and providing global consistency.
>
> **Filter unnecessary nodes to be aggregated**: InterpGCN-GW requires aggregating a significantly smaller number of nodes compared to graph transformers. Graph transformers with node-pair attention tend to aggregate information from all nodes, potentially incorporating noise and diminishing the impact of important nodes. Our method selectively stores important node embeddings in the memory through attention, effectively acting as a condensation or filter of nodes.
>
>
> [9] Haddouche M, Guedj B. Online pac-bayes learning. NeurIPS 2022.
>
> [10] Sharma A, Veer S et al. PAC-Bayes Generalization Certificates for Learned Inductive Conformal Prediction. NeurIPS 2023.
>
> [11] Arezou Rezazadeh. A Unified View on PAC-Bayes Bounds for Meta-Learning. ICML 2022
>
> [12] Haotian Ju et al. Generalization in Graph Neural Networks: Improved PAC-Bayesian Bounds on
> Graph Diffusion. AISTATS 2023.
>
> [13] Alhabib Abbas et al. PAC-Bayesian Bounds on Rate-Efficient Classifiers. ICML 2022.
>
>
> ---
> We hope that our responses address the reviewer's concerns and look forward to further discussions to improve our work.

---

> ### Comment · Reviewer_kZmE · 2023-11-22
> **Responses to the Authors**
>
> Thanks the authors for their detail response. The additional experimental results make the results more convinced. I have raised my score to 6.

---

> > ### Author Response · Authors · 2023-11-22
> >
> > Thank you for acknowledging the additional experimental results and for raising your score. We greatly appreciate your constructive feedback, which has been invaluable in strengthening our paper. Your support and recognition are very encouraging.

---

### Official Review · Reviewer_wkbK · 2023-11-05

**Soundness:** 3 good
**Presentation:** 2 fair
**Contribution:** 2 fair
**Rating:** 6
**Confidence:** 3

**Summary:**

This paper focuses on understanding and improving the generalization ability of Transductive Graph Neural Networks (GNNs) by enhancing the interplay between training and test nodes. The authors address the problem of information flow shortage between distant nodes and intra-batch nodes in GNNs, which hampers their generalization ability. They propose incorporating the L-hop interplay for an L-layer GNN in a PAC-Bayesian bound to quantitatively analyze the influence of the interaction between training and testing sets on generalization ability. Additionally, they introduce a plug-and-play module called InterpGNN-GW, which enhances the interplay between distant and intra-batch nodes using a Graph Global Workspace and key-value attention mechanism. The authors validate their theory and approaches through experiments on small-scale and large-scale graph datasets.

**Strengths:**

Solid theoretical analysis and interesting idea.

**Weaknesses:**

See below

**Questions:**

1. "results concerning graph structure is missing". Homophily/Heterophily is one of the most important properties of the graph structure, but is missed in this paper[1-4].
See [4] for a study with PAC-Bayesian generalization bound.
2. "more information interaction between test and training nodes can lead to smaller generalization error" I doubt the correctness of this claim on heterophilic graphs, see [3].
3. Compared to the graph transformers based on inner-batch pair-wise attention, the global workspace can provide global consistency." The idea of building global consistency is interesting.
4. "Implicit connection" [5] estabilsh graph rewiring method to rebuild some implicit connection between nodes.
5. "the labels of training set $S_Y$ , the features of both training and test nodes $X = [S_X,U_X]$". The notations are wierd. I suggest "$S_Y$" --> "$Y_S$", "$X = [S_X,U_X]$" --> "$X = [X_S,X_U]$".
6. "premise that embedding exchange among connected nodes leads to closer output logits" This premise is invalid for heterophilic graphs. See examples in [1-3].
7. "Theorem 1 interpret the structure imbalance phenomenon." How does theorem 1 interpret the structure imbalance phenomenon? You need to elaborate it in the main paper. You only test on homophilic datasets (Cora, CiteSeer, PubMed). To make claims about grouping effect, you need also test on heterophilic datasets.
8. Ablation study on memory and the gate control of memory doesn't show the effectiveness of memory and the gate control. The proposed models doesn't show statistically better performance than SOTA. Again, the model needs to be tested on heterophilic graphs.

Although missing lots of related works and analysis on heterophilic graphs, I still find this paper interesting. I will consider raising my score if the authors address my concerns well.


[1] Is Homophily a Necessity for Graph Neural Networks?. In International Conference on Learning Representations 2022.

[2] Revisiting heterophily for graph neural networks. Advances in neural information processing systems, 35, 1362-1375.

[3] When do graph neural networks help with node classification? Investigating the impact of homophily principle on node distinguishability. arXiv preprint arXiv:2304.14274.

[4] Demystifying Structural Disparity in Graph Neural Networks: Can One Size Fit All?. arXiv preprint arXiv:2306.01323.

[5] Understanding over-squashing and bottlenecks on graphs via curvature. In International Conference on Learning Representations, 2021.

---

> ### Author Response · Authors · 2023-11-18
> **Response to Reviewer wkbK (Part 1)**
>
> We deeply appreciate your acknowledgment of the solid theoretical analysis and the interesting idea presented in our paper. We also thank the reviewer for highlighting the significance of homophily and heterophily in graph structures. Although our study primarily investigates homophily graphs, aligning with prevalent research trends, both the theoretical foundations and the methods developed in our work have applicability to heterophily graphs. Theoretically, our approach can be expanded to include heterophily assumptions, thereby extending its relevance (See R1). Moreover, the global workspace designed to enhance non-neighbor node interplay effectively in both homophily and heterophily graphs (see R2 and R8). This dual applicability reflects the versatility and robustness of our approach in addressing diverse graph structures. Below is our detailed response.
>
> **Q1:  The disscussion of hetyerophily graph is missing. See [4] for a study with PAC-Bayesian generalization bound.**
>
> **A1**:  Thank you for pointing out the need to discuss heterophily graphs. Theorem 1 in our work can be extended to include heterophily graphs with an additional assumption. The primary distinction between homophily and heterophily graphs lies in whether connected nodes belong to the same category. In our theoretical derivation, we assume that the true labels are solely dependent on features, meaning nodes with similar features are likely to be in the same class. Formally, for two nodes $i$ and $j$ with labels $k$, the difference in their label probabilities is bounded by their feature difference:
>
> $$ |P(y_i=k|x_i) - P(y_j=k|x_j)| \leq C|x_i - x_j|, $$
>
> where $C$ is a constant determined by the true distribution. In heterophily graphs, we can add a heterophily coefficient related to node distance to this assumption:
>
> $$ |P(y_i=k|x_i) - P(y_j=k|x_j)| \leq C \cdot h_r^{(uv)} \cdot |x_u - x_v|, $$
>
> where $r$ is the distance between nodes $i$ and $j$. In homophily graphs, $h_1$ is small, indicating a tendency for neighboring nodes to be in the same class. Conversely, in heterophily graphs, $h_1$ is larger, implying neighboring nodes are likely to belong to different classes. Similar to our defined L-hop interplay between training set $S$ and test set $U$, we can define L-hop heterophily as:
>
> $$ \mathcal{H}_ L  = \frac{ \sum_{r=1}^L (L-r+1) \cdot |h_r^{i,j}|}{|S| \cdot |U|}, $$
>
> where $i \in S$, $j \in U$. In the final generalization error, an additional term on $\mathcal{H}_L$ is included. Taking the dating network dataset from [6] as an example, adjacent nodes are more likely to belong to different classes. Here, due to typically small interplay values, L-hop heterophily $\mathcal{H}_L$ has a more significant impact on generalization error than L-hop interplay $(1-I_L)$. For non-adjacent nodes, as heterophily is usually not strong (as discussed in R2), L-hop interplay remains the dominant factor.
>
>
> Regarding the PAC-Bayesian generalization bound in [4], a similar approach to heterophily is introduced. However, [4] makes strong assumptions about graph datasets, limiting the extendibility of the derived generalization error bounds. Their generalization error is based on that graph is generated with a contextual stochastic block model (CBSM), assuming features of different classes originate from distinct normal distributions, with different connection probabilities $p$ and $q$ for inter-class and intra-class nodes, respectively. The final generalization error bound based on [7] includes the homophily ratio difference between training and test sets.

---

> ### Author Response · Authors · 2023-11-18
> **Response to Reviewer wkbK (Part 2)**
>
> **Q2: The correctness of “more information interaction between test and training nodes can lead to smaller generalization error”.**
>
> **A2**: The statement regarding the impact of information interaction on generalization error can be more accurately phrased as follows: "More information interaction between neighboring test and training nodes can lead to a smaller generalization error for homophily graphs, while more information interaction between non-neighboring test and training nodes can lead to a smaller generalization error for heterophily graphs." This concept is supported by a survey on heterophily GNNs, which states, "On heterophilic graphs, nodes with high structural and semantic similarities might be farther away from each other," and "The representation ability of heterophilic GNNs can be significantly improved by capturing important features from distant but informative nodes," as quoted from [8].
>
>
> These elucidate why our method  by writing important nodes into the memory and broadcasting them to all nodes, exhibits superior performance on heterophily graphs (experimental results in R8).  Similarly, many GNNs aggregate information from non-neighboring neighbors. For instance,  MixHop [9], H2GCN [10] propose aggregating information from higher-order neighbors at each message passing step. Similarly, UGCN [11] utilizes two-hop networks for message passing, while GGCN [12] employs cosine similarity to send signed neighbor features under constraints of relative node degrees. GPR-GNN [13] uses learnable weights to adaptively combine representations from each layer via the Generalized PageRank (GPR) technique.
>
> The discussion in [3] does not conflict with these statements.  [3] still focuses on the influence of neighboring nodes' label distribution, which necessitates considering the inter-class distinguishability of neighbors.  In the example provided in [3], if two nodes have 2 and 4 neighboring nodes with labels (2, 3) and (2, 2, 3, 3), they would still be classified similarly after aggregation. This supports the necessity of aggregating non-neighboring node features, consistent with our method's approach.
>
>
> **Q3: The idea of building global consistency is interesting.**
>
> **A3**: We appreciate your recognition of the capability of our methods for ensuring global consistency.
>
> **Q4: "Implicit connection" [5] estabilsh graph rewiring method to rebuild some implicit connection between nodes.**
>
> **R4**: We will include the graph rewiring method in our related works section.
>
> **Q5: The notations of $S_x$ and $S_Y$ are wierd.**
>
> **A5**: Thank you for the feedback on notations; we will make the suggested corrections in our revised version.
>
>
> **Q6: "premise that embedding exchange among connected nodes leads to closer output logits” The correctness of heterophilc graphs.**
>
> **A6**: To clarify, the output logits referred to are the outputs of the final layer of the graph neural network. Consider a simple graph neural network with only two nodes, $i$ and $j$. In the case where $i$ and $j$ are not connected, the difference in their outputs is $|\sigma(x_iW) - \sigma(x_jW)|$, where $\sigma$ denotes the activation function, and $ W $ is the weight matrix. However, if $i$ and $j$ are connected, their  output after embedding exchange becomes $|\sigma\left(\frac{x_u+x_v}{2}W\right) - \sigma\left(\frac{x_u+x_v}{2}W\right)|$, which simplifies to 0. This demonstrates that embedding exchange among connected nodes indeed leads to closer output logits. In the context of heterophilc graphs, we only need to refine our thorem to reflect that closer outputs among neighboring nodes can lead to larger generalization error on heterophilc graphs, as indicated in **A1**.

---

> ### Author Response · Authors · 2023-11-18
> **Response to Reviewer wkbK (Part 3)**
>
> **Q7: How does Theorem 1 interpret the sturcture imbalance phenomenon. Need for test of grouping effect on heterophilc dataset.**
>
> **A7** If the test set nodes have more connections with the training set nodes, this results in a larger $L$-hop interplay value, leading to a smaller generalization bound value. For instance, consider two groups selected from a dataset: Group 1 ($U_1$) comprising nodes directly adjacent to the training set, and Group 2 ($U_2$) consisting of nodes with no pathways to the training set. In a three-layer graph neural network, the 3-hop interplay for $U_1$, denoted as $\mathcal{I}_2^{(1)}$, will be at least $\frac{2}{|U_1|}$. In contrast, the 2-hop interplay for Group 2 will be 0. Therefore, the generalization error upper bound for Group 1 is smaller than that for Group 2.
>
> We conducted grouping tests using GCN on heterophilic datasets including Actor, Chameleon, and Squirrel. We select 20 nodes from each class for the training set, 500 for validation set, and 1000 for test nodes. After repeating the experiments 100 times, the average results are as follows:
>
> | Group index | Actor | Chameleon  | Squirrel |
> | --- | --- | --- | --- |
> | 1 | 22.3 | 51.5 | 33.3 |
> | 2 | 21.5 | 47.4 | 32.0 |
> | 3 | 22.6 | 45.9 | 31.0 |
> | 4 | 22.6 | 45.1 | 32.2 |
> | 5 | 23.2 | 39.4 | 28.1 |
>
> As observed, in Chameleon and Squirrel datasets, different groups show a tendency to decline in performance correlating with the decrease in interplay.
>
> **Q8: Ablation study on memory and the gate control of memory doesn't show the effectiveness of memory and the gate control. The proposed models doesn't show statistically better performance than SOTA. Again, the model needs to be tested on heterophilic graphs.**
>
> **A8**: In response to the reviewer's comments regarding the ablation study on memory and gate control, as well as the performance comparison with state-of-the-art (SOTA) models, especially on heterophilic graphs, our clarifications and findings are as follows:
>
> 1. Firstly, it's important to clarify that our ablation study of Memory is not about removing the memory part but initializing it at each training epoch. Memory still can  store node features. A performance drop is observed if we completely remove the memory part, i.e., compare ClusterGCN with InterpGCN-GW. The gate control aspect follows the implementation from previous global workspace studies, and we agree that its contribution to the graph global workspace is relatively small.
>
> 2. Our method shows modest improvements on small-scale graphs but large benefits on large-scale graphs. This advantage stems from the ability of our global workspace component to facilitate information exchange with nodes in other batches during the training and test phase. Following Reviewer mAhi's suggestion, we also tested our model in a non-rigorous inductive learning setting, where the model does not access test node features during training, yet our methods can broadcast information from the training phase to test nodes. In this scenario, other models exhibit a substantial performance drop compared to transductive learning, while our method achieves results similar to those in the transductive setting. For a detailed presentation of these results, please refer to **A5** to Reviewer mAhi.
>
>
>
>
>
> 3. We have conducted experiments on heterophily graph datasets, using official Github implementations with default hyperparameters for the baselines. The results are as follows:
>
>
> |                 | Actor          | Squirrel        | Chameleon      |
> | --------------- | -------------- | --------------- | -------------- |
> | GCN             | 28.92 $\pm$ 1.67 | 57.21 $\pm$ 1.66  | 62.12 $\pm$ 1.74 |
> | NodeFormer      | 29.93 $\pm$ 1.78 | 49.12  $\pm$ 1.22 | 58.73 $\pm$ 1.84 |
> | InterpGCN-GW    | 31.57 $\pm$ 1.29 | 60.12  $\pm$ 1.31 | 64.27 $\pm$ 1.96 |
> | GPRGNN [13]          | 33.32 $\pm$ 1.71 | 46.20 $\pm$ 0.76  | 51.71 $\pm$ 2.49 |
> | GGCN [12]            | 36.79 $\pm$ 1.97 | 52.19 $\pm$ 1.41  | 66.19 $\pm$ 2.73 |
> | H2GCN [10]          | 35.21 $\pm$ 2.45 | 40.79 $\pm$ 1.42  | 60.13 $\pm$ 1.78 |
> | InterpGPRGNN-GW | 36.55 $\pm$ 1.79 | 45.73 $\pm$ 1.15  | 52.19 $\pm$ 2.05 |
> | InterpGGCN-GW   | 37.81 $\pm$ 2.17 | 55.49 $\pm$ 1.92  | 67.17 $\pm $1.54 |
> | InterpH2GCN-GW  | 35.42 $\pm$ 1.79 | 41.46 $\pm$ 1.83  | 59.79$ \pm$ 1.97 |

---

> ### Author Response · Authors · 2023-11-18
> **Response to Reviewer wkbK (Part 4 for reference)**
>
> [6] Zhu J, Rossi R A, Rao A, et al. Graph neural networks with heterophily. AAAI 2021.
>
> [7] Ma et al, Subgroup generalization and fairness of graph neural networks. NeurIPS 2021.
>
> [8] Zheng X, Liu Y, Pan S, et al. Graph neural networks for graphs with heterophily: A survey. arXiv preprint arXiv:2202.07082, 2022.
>
> [9] Abu-El-Haija S, Perozzi B, Kapoor A, et al. Mixhop: Higher-order graph convolutional architectures via sparsified neighborhood mixing. ICML 2019.
>
> [10] Zhu J, Yan Y, Zhao L, et al. Beyond homophily in graph neural networks: Current limitations and effective designs. NeurIPS 2020.
>
> [11] Jin D, Yu Z, Huo C, et al. Universal graph convolutional networks. NeurIPS 2021.
>
> [12]  Wang T, Jin D, Wang R, et al. Powerful graph convolutional networks with adaptive propagation mechanism for homophily and heterophily. AAAI 2022.
>
> [13] Chien E, Peng J, Li P, et al. Adaptive universal generalized pagerank graph neural network. ICLR 2020.
>
> ---
> We hope that our responses adequately address your concerns and look forward to any further feedback you may have.

---

> ### Author Response · Authors · 2023-11-22
>
> Thank you once again for your thorough and insightful feedback. We have endeavored to address all your concerns in our responses.  As we near the end of this discussion phase, we are eager to know if our explanations have satisfactorily addressed your points.
>
> If you have any more comments or questions about our rebuttal,  we strongly welcome your feedback. Your guidance is crucial for improving our work, and we look forward to any additional thoughts you might have.

---

> > ### Comment · Reviewer_wkbK · 2023-11-23
> >
> > Thanks for the response. The authors have addressed most of my concerns and I'll raise my rating to 6. However, when I checked the revised paper and appendix, I didn't find the discussion on heterophilic graphs. And also, I suggest comparison with the SOTA models on heterophilic graphs, e.g. [1,2].
> >
> > [1]  Revisiting heterophily for graph neural networks. Advances in neural information processing systems, 35, 1362-1375.
> > [2] Neural sheaf diffusion: A topological perspective on heterophily and oversmoothing in gnns. NeurIPS, 2022.

---

> > > ### Author Response · Authors · 2023-11-23
> > >
> > > We express our sincere gratitude for your acknowledgment of our response and for raising the rating. We would like to clarify that in the previous version, experiments with discussions for InterpGNN on heterophilic graphs were indeed added in Section 5.4. In the newly updated version, we have made the following addtions:
> > >
> > > 1. In Appendix G.4, we expand the discussion on the scalability of our theoretical analysis to heterophilic graphs (A1 and A7 in initial response).
> > > 2. In Appendix G.4, we also inlude a  discussion on why our method is apt for heterophilic graphs (A2 and A8 in initial response).
> > > 3. We have refined the statement in your Q2 to limit it to homophilic graphs.
> > > 4. All reference you mentioned have been added, and we also compare with STOA baselines as per your suggestion.
> > >
> > > Once again, we greatly appreciate your constructive feedback and the opportunity to enhance our manuscript further.

---

> ### Author Response · Authors · 2023-11-23
>
> Dear reviewer, thank you very much for your suggestion to focus on heterophilic graphs. We have made every effort to address your concerns, including the analysis and experiments on heterophilic graphs. Your initial comments, noting our work's solid theoretical analysis and interesting ideas, have greatly motivated us. Your insights have been a guiding force in our revisions, and we are hopeful that these updates meet your expectations.
>
> As we approach the end of the discussion phase with less than 12 hours remaining, we eagerly await your further insights on our revisions. Your final evaluation is very important to us.

---

### Official Review · Reviewer_mAhi · 2023-11-08

**Soundness:** 2 fair
**Presentation:** 3 good
**Contribution:** 3 good
**Rating:** 6
**Confidence:** 3

**Summary:**

This paper aims to propose a theory to derive a PAC-Bayesian bound for message-passing GNN in the transductive node classification and interpret the structure imbalance phenomenon. Based on the theory, authors propose the InterpGNN-GW to enhance the interplay between distant and intra-batch nodes via an attention-based global workspace mechanism.

**Strengths:**

1. The derived bound in this paper shows that more information interaction between test and training nodes can lead to smaller generalization errors and interpret the structural imbalance phenomenon.
2. To address the problem, the authors proposed a new method called InterpGNN-GW which leverages the attention mechanism to global workspace.
3. Sufficient experiments on different graph datasets have been conducted. The proposed InterpGNN-GW outperforms many baseline methods both on small-scale graphs and large-scale graphs.
4. The paper presentation is good and the organization is clear and easy to follow.

**Weaknesses:**

1. In the ‘Theorem 1 interpret the structure imbalance phenomenon.’ part, authors claim that stronger interplay with training nodes brings small generalization error, and nodes with smaller degrees engage in fewer interactions with training nodes thus having the trend in Fig2. However, it seems like a mistake according to the authors' theory. Nodes with smaller degrees engage in fewer interactions with training nodes should have bigger generalization error, while in fig2 nodes with fewer degrees have better performance.
2. The Experiment part mainly concentrates on node classification performance and ablation study. It would be better to design experiments to show how InterpGNN-GW increases the interplay between training and test nodes, which also strengthens the claims in theory and interprets the structure imbalance phenomenon.
3. Lack the motivation to concatenate positional encoding. Authors may add motivation such as position embedding containing the interplay with training and test nodes.
4. It would be more convincing if authors could provide full ablation study results on all datasets instead of separating ablated models to Tab 5 and Fig 6 and using different datasets.
5. Can InterpGNN-GW be used in a non-rigorous inductive setting? If yes, what about performance? For example, train InterpGNN-GW on the training nodes and broadcast the workspace to test nodes with test nodes still unseen to the training process. It should be similar to interplaying among out-of-batch nodes mentioned in the intuition part.
6. Why does InterpGNN-GW fail to outperform some baselines on CoraFull, any in-depth or intuitive analysis?

**Questions:**

See Weaknesses

---

> ### Author Response · Authors · 2023-11-18
> **Response to Reviewer mAhi (Part 1)**
>
> We are grateful for your insightful comments and positive feedback on our paper. Below are our detailed response.
>
> **Q1:  Nodes with smaller degrees engage in fewer interactions with training nodes have bigger generlization error.**
>
> **R1**: We apologize for any ambiguity in Figure 2. The x-axis represents group indexes, not node degrees. Nodes are sorted by degree and divided into five groups, with Group 1 (lowest index) having the highest node degrees and Group 5 (highest index) the lowest.
>
> **Q2: How interpGNN-GW increase the interplay between training and test nodes.**
>
> **R2**:  To demonstrate how InterpGCN-GW enhances the interplay between training and test nodes, we present a specific example with a 3-layer GCN.  We select a training set of $N_s$ nodes and $N_n$ test nodes isolated from the training set, where the initial L-hop interplay is 0. InterpGCN-GW, through attention mechanism, stores information about important nodes, essentially holding a mix of various node embeddings in  memory. For analytical purposes, we assume that the memory contains the full features of $N_m$ training nodes, which can be broadcasted to all test nodes. This effectively transforms all test nodes into 1-hop neighbors of training nodes of training nodes, connected to at least $N_m$ training nodes. Consequently, the interplay increases to at least $\frac{6}{N_m}$, leading in a smaller generalization error bound.
>
> **Q3: Motivate to concatenate positional encoding.**
>
> **R3**: The concatenation of node2vec positional embedding in our method is driven by two main considerations:
>
> **Topology Information**: Positional embedding encapsulates a node's topological information within the entire graph, meaning nodes that are closer in the graph have similar positional embeddings. Since large-scale GNNs typically use mini-batch training, nodes in different batches are isolated. By reading and writing node features to the workspace, positional embedding informs about the relative position of nodes in different batches within the entire graph. If a node's positional embedding in the workspace closely matches that of a test node, it indicates proximity in the graph, thus it's more likely to be relevant for updating the test node's embedding.
>
> **Enhanced Expressiveness**: Concatenating positional embedding, as opposed to adding it, enhances the expressive power of GNNs in graph isomorphism tests. As demonstrated in [1], increasing feature distinctiveness (by adding random features) significantly boosts GNN performance. This is because it helps distinguish node positions within the entire graph. Concatenation [X, X_pe] provides more variance compared to addition (X+X_pe), thereby increasing the network's expressiveness.
>
> [1] Sato R, Yamada M, Kashima H. Random features strengthen graph neural networks. SDM 2021.

---

> > ### Comment · Reviewer_mAhi · 2023-11-21
> > **Thanks for your clarification**
> >
> > Thank you for providing a detailed clarification. However, I still have some concerns regarding Question 2:
> >
> > 1. I recommend that the authors consider designing experiments to illustrate how InterpGNN-GW functions by enhancing the interplay. It would be more beneficial to present concrete experimental results rather than relying solely on a thought experiment. For instance, visualizing how node representations change after interplaying or investigating which types of nodes in the workspace are chosen for interplay with corresponding test nodes (e.g., nodes with the same label?) could provide valuable insights. This is just a suggestion, and you are encouraged to design other experiments that demonstrate the in-depth analysis of the interplay mechanism.
> >
> > 2. In addition to the experiments, the theoretical aspect appears somewhat decoupled from InterpGNN. While the theory could explain the motivation, it may lack a stronger connection between the theory and the InterpGNN model itself.

---

> ### Author Response · Authors · 2023-11-18
> **Response to Reviewer mAhi (Part 2)**
>
> **Q4: Full ablation study on all datsets.**
>
> **R4**: We revise the paper to include a unified ablation study on 2 small graphs and all 3 large graph datasets.
>
> **Q5:  Can InterpGNN-GW be used in a non-rigorous inductive setting? If yes, what about performance? For example, train InterpGNN-GW on the training nodes and broadcast the workspace to test nodes with test nodes still unseen to the training process. It should be similar to interplaying among out-of-batch nodes mentioned in the intuition part.**
>
> **R5**: Thank you for your valuable insight, which has highlighted an advantage of InterpGCN-GW which we did not noticed. Unlike other methods that experience significant performance drops in inductive settings compared to transductive settings, InterpGCN-GW shows only a minor decrease in performance in non-rigorous inductive settings, closely aligning with its transductive results. In scenarios where test node features are unseen during training, the performance of InterpGCN-GW is as follows:
>
> | Model         | Amazon2M         | Aminer-CS        | Reddit  |
> |---------------|------------------|------------------|------------------|
> | ClusterGCN    | 83.85 $\pm$ 0.27 | 61.59 $\pm$ 0.37 | 89.77 $\pm$ 0.35 |
> | GRAND+        | 84.12 $\pm$ 0.18 | 62.48 $\pm$ 0.36 | 92.05 $\pm$ 0.23 |
> | NodeFormer   | 84.25 $\pm$ 0.41 | 62.12 $\pm$ 0.24 | 90.75 $\pm$ 0.25 |
> | InterpGCN-GW  | 88.13 $\pm$ 0.24 | 67.85 $\pm$ 0.19 | 92.79 $\pm$ 0.19 |
>
> **Q6: Why does InterpGNN-GW fail to outperform some baselines on CoraFull, any in-depth or intuitive analysis?**
>
> **R6**: InterpGCN-GW only fail to outperform GRAND+ with small difference on CoraFull dataset. This can be analyzed based on several dataset-specific factors:
>
> **High Feature Dimensionality**: CoraFull has a feature dimensionality of 8710, far exceeding that of datasets like WikiCS (300), Computer (767), Reddit (602), and Amazon2M (100).
>
> **Feature Sparsity**: The dataset's features are one-hot vectors and highly sparse. This, combined with the high dimensionality, predisposes models like InterpGCN-GW, which employ attention mechanisms, to potential overfitting.
>
> **Method of Sparsification**: GRAND+, in contrast, uses Topk sparsification, effectively mitigating overfitting in high-dimensional, sparse scenarios.
>
> **Heterophily of Corafull**: CoraFull tends more towards heterophily, with lower proportions of connected nodes sharing similar labels compared to other datasets.
>
> ---
> Thank you again for your constructive observations and appreciation of our work.

---

> ### Author Response · Authors · 2023-11-21
> **Further response to remaining concerns**
>
> Thank you once again for your valuable suggestions, which guide us in enhancing our paper. Here's our response to your remaining concerns:
>
> A1.  We follow your advice and visualize the attention scores to elucidate the function of InterpGNN in enhancing interplay. We select 40 nodes within the Cora dataset for a clear comparison, using 32 memory slots in our model. These newly-added figures are detailed in Appendix G.3. Figure 8 illustrates the second step’s attention over nodes, indicating preferences for nodes to be stored in memory. Figure 9 visualizes the third step’s attention over memory, revealing nodes’ preferences in reloading information from memory slots. Figure 10 presents the cosine similarity of node features with heat map and annotates shortest path lengths between node pairs. Our key observations include:
>
> 1. Node 34, isolated from other nodes (as seen in Figure 9), cannot engage in message passing in GCN. In our approach, Node 34 accesses information from other nodes via memory slots.
> 2.  Nodes 27 and 16 share high feature similarity and the same label, but are separated by a shortest path length of 9, which impedes direct message passing in a Vanilla GCN. In our approach, Node 27's features are written into the memory, and Node 16 can access this information by reading from the memory. This allows for effective communication between these two nodes despite their distance in the graph structure.
> 3.  Memory tends to store information from Node 26 and 27, maybe because they share relatively high feature similarity with part of nodes, marking them as  important nodes.
>
> A2. We derive a PAC-Bayesian generalization bound for GCN equipped with global workspace in Appendix G.3, which is a simplification of InterpGNN by removing attention. Our Theorem 1 originally aims to analyze the impact of graph data structure on message-passing GNNs. To bridge the gap you mention, we add a corollary in Appendix G.3, which is the PAC-Bayesian generalization error of GCN with global workspace (based on the process in our initial response A2 to your question 2).  This shows that adding a global workspace (without attention) enhances interplay between nodes, thus reducing the generalization error upper bound of GCN.
> This finding can be supported with [1], which use dummy nodes connected to all nodes to boost graph structure learning, since memory slots without attention can act as virtual nodes. Building on this corollary, we use attention to selectively write important node information to memory.
>
> Extending Theorem 1 to fully encompass InterpGNN is challenging, as the selection of nodes for memory is real-data-dependent. Furthermore, Theorem 1 factors in model complexity. The introduction of an attention mechanism in InterpGNN could potentially lead to a higher PAC-Bayesian generalization error compared to GCN. Thus, we empirically validate that InterpGNN enhances interplay, as discussed in the first paragraph. Nevertheless, exploring PAC-Bayesian analysis for InterpGNN and integrating positional embedding into Theorem 1 are our future directions, as you suggest.
>
> [1] Liu, Xin, et al. Boosting graph structure learning with dummy nodes. ICML 2022.
>
> We hope this response can addrese your remaining concers. We are open for furthrer discussuon with  you which is really inspirational.

---

> > ### Comment · Reviewer_mAhi · 2023-11-22
> > **Thank you for the clarifications**
> >
> > Thank the authors for the further clarifications and convincing experiments! All my concerns are addressed and I would like to keep my positive rating.

---

> ### Author Response · Authors · 2023-11-23
> **Thank you for your continued support**
>
> We are deeply grateful for your positive remarks and acknowledgment of our clarifications and experiments. Thank you for your continued support and valuable feedback throughout this process!

---

### Comment · Area_Chair_fnMw · 2023-11-10
**Authors-Reviewers discussion starts today, ends on Nov 22**

Dear authors and reviewers,

@Authors: please make sure you make the most of this phase, as you have the opportunity to clarify any misunderstanding from reviewers on your work. Please write rebuttals to reviews where appropriate, and the earlier the better as the current phase ends on Nov 22, so you might want to leave a few days to reviewers to acknowledge your rebuttal. After this date, you will no longer be able to engage with reviewers. I will lead a discussion with reviewers to reach a consensus decision and make a recommendation for your submission.

@Reviewers: please make sure you read other reviews, and the authors' rebuttals when they write one. Please update your reviews where appropriate, and explain so to authors if you decide to change your score (positively or negatively). Please do your best to engage with authors during this critical phase of the reviewing process.

This phase ends on November 22nd.

Your AC

---

### Author Response · Authors · 2023-11-20
**General Response**

We sincerely appreciate the time and effort invested by the reviewers in providing comprehensive and insightful feedback on our submission. The consensus among all five reviewers is that our derived generalization error is a strong aspect of the paper, described as “solid” (wkbK), “interesting” (kZmE, qP57), and a “noteworthy contribution” (fk5N). Apart from a possible misunderstanding by one reviewer, the remaining four reviewers recognize the novelty and effectiveness of our method, which is termed “new” (mAhi), “interesting” (wkbK), “reasonable” (kZmE), and “offering significant advantage” (fk5N). They also acknowledge our experimental work, which they find “sufficient” (mAhi, kZmE), and “well thought out” (fk5N).

During the author response period, we have undertaken every effort to address the concerns raised by the reviewers and to verify our claims with additional experiments. Our key responses are summarized as follows:
1. Theoretical clarifications and extensions:

    i) We elucidate why enhancing interplay can improve generalization error and how InterpGNN specifically augments interplay. (A2 to mAhi, A3 to kZmE)

    ii) We discuss the applicability of our theory to heterophily graphs (A1 to wkbK).

    iii) We justify the rationale behind assuming randomness originates from labels and corroborated this with experimental validation on a synthetic graph (A1 to kZmE).

2. Method refinements and experimental validation:

    i) We demonstrate the applicability of our method to heterophily graphs, supported by experimental evidence (A2, A8 to wkbK).

    ii) We further validate the advantages of our method in a non-rigorous inductive setting (A5 to mAhi).

    iii) We provide a motivation for the concatenation of positional embedding (A3 to mAhi).

3. Enhancements to presentation:

    i) We clarify the axis and grouping rules of Figs. 1 and 2 in response to queries.

    ii) We complete ablation study results on all datasets, add standard deviations, and correct and clarify certain equations.

In our individual reponse to each reviewer, we have carefully addressed each specific questions and concerns. We  hope that our response could resolve reviewers' concerns and look forward the further replies from AC and reviewers.

---

### Meta-Review · Area_Chair_fnMw · 2023-12-05

**Metareview:**

This meta-review is a reflection of the reviews, rebuttals, discussions with reviewers and/or authors, and calibration with my senior area chair. This paper investigates the generalisation ability of transductive graph neural networks and introduces new PAC-Bayes generalisation bounds. There is a consensus among reviewers that the paper contributes a solid theoretical analysis. Several reviewers have raised their scores after the rebuttals, which clarified some of the concerns expressed in reviews.

**Justification For Why Not Higher Score:**

Good evaluation but there is a lack of enthusiasm for going higher.

**Justification For Why Not Lower Score:**

The paper contributes an important theoretical analysis and deserves to be featured at ICLR in my opinion.

---

### Decision · Program_Chairs · 2024-01-16

Accept (poster)